# SYSTEM-1.$x$: LEARNING TO BALANCE FAST AND SLOW PLANNING WITH LANGUAGE MODELS

**Swarnadeep Saha**          **Archiki Prasad**          **Justin Chih-Yao Chen**

**Peter Hase**          **Elias Stengel-Eskin**          **Mohit Bansal**

UNC Chapel Hill

## ABSTRACT

Language models can be used to solve long-horizon planning problems in two distinct modes. In a *fast* 'System-1' mode, models directly generate plans without any explicit search or backtracking, and in a *slow* 'System-2' mode, they plan step-by-step by explicitly searching over possible actions. System-2 planning, while typically more effective, is also computationally more expensive and often infeasible for long plans or large action spaces. Moreover, isolated System-1 or System-2 planning ignores the user's end goals and constraints (e.g., token budget), failing to provide ways for the user to control the model's behavior. To this end, we propose the *System-1.x Planner*, a framework for controllable planning with language models that is capable of generating hybrid plans and balancing between the two planning modes based on the difficulty of the problem at hand. System-1.$x$ consists of (i) a controller, (ii) a System-1 Planner, and (iii) a System-2 Planner. Based on a user-specified hybridization factor $x$ governing the degree to which the system uses System-1 vs. System-2, the controller decomposes a planning problem into sub-goals, and classifies them as easy or hard to be solved by either System-1 or System-2, respectively. We fine-tune all three components on top of a single LLM, requiring only search traces as supervision. Experiments with two diverse planning tasks – Maze Navigation and Blocksworld – show that our System-1.$x$ Planner outperforms a System-1 Planner, a System-2 Planner trained to approximate A$^*$ search, and also a symbolic planner (A$^*$ search), given a state exploration budget. We also demonstrate the following key properties of our planner: (1) *controllability*: by adjusting the hybridization factor $x$ (e.g., System-1.75 vs. System-1.5) we can perform more (or less) search, improving performance, (2) *flexibility*: by building a neuro-symbolic variant composed of a neural System-1 planner and a symbolic System-2 planner, we can take advantage of existing symbolic methods, and (3) *generalizability*: by learning from different search algorithms (BFS, DFS, A$^*$), we show that our method is robust to the choice of search algorithm used for training.[1]

## 1 INTRODUCTION

A key feature of intelligence – both human and artificial – is the ability to plan. Indeed, planning has been a major topic of AI research for most of its history (Russell and Norvig, 2016), with applications ranging from navigation and robotics to manufacturing and story-telling. These days, auto-regressive Large Language Models (LLMs) are increasingly used for various long-horizon planning and reasoning tasks (Bubeck et al., 2023; Wei et al., 2022; Yao et al., 2024). Despite some initial promise, recent careful investigations have highlighted LLMs' shortcomings on classical planning tasks (Pallagani et al., 2023; Valmeekam et al., 2023; 2024; Momennejad et al., 2024). Supervising models with correct plans has also led to limited success, especially for out-of-distribution generalization (Dziri et al., 2023). Fundamentally, this issue is attributed to a lack of explicit search, backtracking, or learning from mistakes (Kambhampati et al., 2024; Bachmann and Nagarajan, 2024; Gandhi et al.,

---

[1]Code available at `https://github.com/swarnaHub/System-1.x`

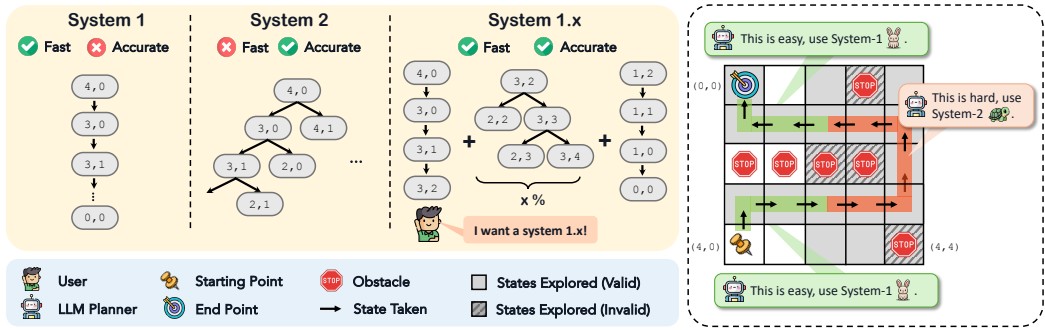

Figure 1: Comparative overview of a System-1 Planner, a System-2 Planner, and our hybrid and controllable System-1.$x$ Planner. Given a Maze Navigation task, a System-1 Planner directly generates a plan as a sequence of steps without search, thus making it fast but inaccurate. A System-2 Planner performs search before generating a plan and hence is more accurate but much slower. Our System-1.$x$ Planner generates hybrid plans, where the level of hybridization is controlled by a user-defined hyperparameter. Hybrid plans balance System-1 sub-plans for easier sub-goals and System-2 sub-plans for harder ones, making System-1.$x$ fast, accurate, and controllable.

2024), all of which are essential components of standard planning systems. LLMs directly generating plans generally resemble humans' *fast* 'System-1' decision-making (Kahneman, 2011), characterized by quick, intuitive judgments rather than slow, deliberate planning (Kambhampati et al., 2024).

As potential remedies, *slow* 'System-2'-inspired LLM planning approaches have emerged that either (1) augment LLMs during inference with symbolic search algorithms (Yao et al., 2024; Sel et al., 2023; Ahn et al., 2022) or (2) directly teach LLMs to carry out search themselves (Gandhi et al., 2024; Lehnert et al., 2024). While the former approach of using external search tools can enhance test-time performance, the latter approach – training models to search – has the advantage of resulting in a single unified search model and also discovering better and more efficient search strategies (Gandhi et al., 2024). However, such training-time approaches also face challenges due to *high computational demands and potential infeasiblity*. For instance, one such recently proposed (trained) System-2 method, Searchformer, uses token sequences that verbalize entire search trajectories (Lehnert et al., 2024). Consequently, during inference, based on the action space and the plan length, these models explore a large number of states, resulting in an increased number of generated tokens. The cost of exploration is further exacerbated by the fact that Transformers (Vaswani et al., 2017) typically exhibit quadratic complexity w.r.t. the sequence length. Thus, as sequence length increases, the computational complexity grows, significantly impacting memory usage and speed, rendering many long-horizon tasks infeasible due to token limits.[2] A more promising alternative would thus be to avoid incurring a high cost on all samples and steps by training a planner with a smarter, more balanced allocation of compute, i.e., one that uses System-1 for easier (sub-)problems while reserving the more expensive System-2 for harder (sub-)problems based on the budget.

To this end, we propose the *System-1.$x$ Planner*, drawing inspiration from the Dual-Process Theory (Wason and Evans, 1974; Kahneman, 2011) that argues for the co-existence of fast and slow planning in humans. The fast system, relying on expertise and quick judgments, can be cheaply and quickly applied either instinctively or in areas where an individual has acquired expertise. The slow system is more costly, involving sequential processing and relying on working memory to formulate a more deliberate plan and includes the consideration of hypothetical alternatives (Evans, 2003). Crucially, humans can switch between these two modes as determined by the task at hand (Kahneman, 2011). Based on this distinction, we develop the System-1.$x$ Planner, a controllable framework for long-horizon planning with LLMs that is capable of generating hybrid plans interleaved with System-1 and System-2 sub-plans. System-1.$x$ offers a promising middle ground between a fast but inaccurate System-1 Planner, used for easy problems, and an accurate but slow System-2 Planner, used for hard problems (see Fig. 1 for an overview).

System-1.$x$ is composed of three trained components: (1) a controller that decomposes a planning problem into easier and harder sub-goals, (2) a System-1 Planner that solves the easier sub-goals

---

[2] For example, full System-2 on $5 \times 5$ mazes with a plan length of 8 requires $\sim 1500$ tokens on average.

without any explicit search or exploration, and (3) a System-2 Planner that solves the harder sub-goals by deliberately searching over possible actions. One of the primary strengths of System-1.$x$ is its *train-time controllability* – based on a user-specified hyperparameter $x \in [0, 1]$, we train a controller that ultimately governs the degree of hybridization between the System-1 and 2 planners. It also offers *test-time controllability*, by which an already trained controller can be adjusted towards solving more or fewer problems using System-2, providing another level of inference-time compute.[3] Importantly, System-1.$x$ is a *fully (LLM-based) neural planner* built on top of one base LLM, using supervision only from search traces and not relying on any external solvers or verifiers. Thus, it has an advantage over inference-time methods that rely on symbolic planners which may not exist for certain domains, making data-driven approaches like System-1.$x$ self-contained and easier to apply to new domains and to scale up. At the same time, our LLM-based System-1.$x$ can also be seamlessly converted into a *neuro-symbolic planner* by replacing the System-2 Planner with a symbolic solver (e.g., A$^*$), allowing us to integrate powerful symbolic tools when they are available.

We conduct experiments on the domains of Maze Navigation and Blocksworld to show that System-1.$x$ Planner matches and generally outperforms a System-1 Planner, a System-2 Planner, and a System-1.$x$ variant (without sub-goal decomposition) at all budgets by generating up to 33% more valid plans. We also report similar findings with our neuro-symbolic System-1.$x$, that in fact typically outperforms symbolic search, beating A$^*$ by up to 39% at a fixed budget of explored states and matching it at maximum budget. Next, we show the controllability of our framework by training a System-1.75 Planner that, compared to a System-1.5 Planner, trades off efficiency for greater accuracy. This trend can be continued to recover the full System-2 performance. System-1.$x$ also generalizes to different search algorithms (BFS, DFS, A$^*$) and exhibits exploration behavior that closely resembles the corresponding algorithm it is trained on. Overall, the System-1.$x$ Planner introduces a novel way of applying LLMs to controllable and hybrid planning, optimizing for both accuracy and efficiency. This, in turn, allows System-1.$x$ to outperform both System-1 and -2 at a fixed budget. By intelligently allocating more resources to harder sub-goals, System-1.$x$ is able to save its System-2 budget for cases where it is necessary.

## 2 System-1.$x$: Controllable and Hybrid Planning with LLMs

### 2.1 Background and Problem Setup

A planning problem can be modeled as a Markov Decision Process $\mathcal{M} = (\mathcal{S}, \mathcal{A}, \mathcal{T}, \mathcal{R})$ defined with a set of states $\mathcal{S}$, a set of actions $\mathcal{A}$ on any given state, a transition function $\mathcal{T} : \mathcal{S} \times \mathcal{A} \to \mathcal{S}$ defining transitions between pairs of states based on an action, and a reward function $\mathcal{R} : \mathcal{S} \to \mathbb{R}$ assigning a reward for reaching the goal state. Given such a planning problem, a start state $s_0 \in \mathcal{S}$ and a goal state $s_g \in \mathcal{S}$, the objective of classical planning is to generate a plan $\mathcal{P} = (a_1, ..., a_n)$, as a sequence of $n$ actions $a_i \in \mathcal{A}$ that helps transition from the start state to the goal state. Using this notation, we define some salient concepts in a planning problem below:

- **System-1 Planner.** We use the term "System-1 Planner" to denote any planner that directly generates a plan as a sequence of actions *without* conducting any explicit exploration or search. System-1 Planners are typically only model-based, trained to only output the final plan. Within the scope of this study, a System-1 Planner will specifically refer to a neural LLM-based planner.

- **System-2 Planner.** We use this term to denote planners that *do* conduct explicit search before generating a plan. Unlike model-based System-1 Planners, System-2 Planners can also be symbolic (e.g., search algorithms like Depth First Search, Breadth First Search, A$^*$). A model-based System-2 Planner, within the scope of this study, will specifically refer to an LLM trained to verbalize search trajectories (defined below) before generating a plan. Broadly, the terms System-1 and System-2 will refer to LLM-based planners while symbolic algorithms will be addressed separately.

- **Search Trajectory.** Given a search algorithm, we define a search trajectory or a search trace as the step-by-step execution of the algorithm for an input problem. In the context of LLMs, this will mean a verbalized representation of a search trajectory. Trajectories consist of the final plan, all intermediate actions explored during search, their corresponding states, their validity, etc.

---

[3]E.g., we can train a System-1.5 model and then add a bias term to the controller at test time to increase the probability of System-2, inducing it to *act* like a System-2 model (or System-1.25, System-1.75, etc.).

- **Valid Plan:** A plan $\mathcal{P} = (a_1, ..., a_n)$ is considered valid if starting from the start state $s_0$, sequentially executing all taken actions in the plan leads to the goal state $s_g$, or more formally, $\mathcal{T}(\mathcal{T}(...\mathcal{T}(s_0, a_1)..., a_{n-1}), a_n) = s_g$. *Plan Validity* is a metric measuring the fraction of valid plans that reach the goal state.

- **#States-Explored:** Given a planning problem and a planner (System-1, 2, or $1.x$), we define #States-Explored ($SE$, in short) as the set of all states (both valid as well as invalid) that are visited by the planner on its way to generating the plan. In the case of an LLM planner, this corresponds to the number of states that the planner needs to verbalize for exploration (see right of Fig. 2 for an example output), and thus adds to the token cost of generation. Since a System-1 Planner generates a plan directly without any additional explorations, it only explores the states that are part of its plan, i.e., $SE_{\text{Sys1}} = n$, where $n$ is the length of the generated plan. A System-2 Planner, on the other hand, will perform additional state explorations before generating a plan, resulting in $SE_{\text{Sys2}} \gg SE_{\text{Sys1}}$. Note that #States-Explored for a System-2 Planner will vary based on the underlying search algorithm or the LLM planner learning from that algorithm's traces. For example, A*, being an informed search algorithm, will typically explore fewer states than Breadth First Search. *Plan Validity* and *#States-Explored* will be our two metrics of interest, using which we will compare the performance at varying budgets of all planners (discussed in §4).

- **Sub-goal.** Finally, planning problems are typically compositional, meaning that they can be decomposed into sub-problems, which we will refer to as sub-goals. Formally, given a plan $P$ between states $s_0$ and $s_g$, a sub-goal can be characterized by a pair of states $(s_i, s_j)$ s.t. $i, j \in [0, g], i \le j$, where $s_i$ and $s_j$ are intermediate states on the plan to get from $s_0$ to $s_g$.

**Problem Setup.** Having defined the key terms above, we now describe the setup underlying our LLM-based System-$1.x$ Planner. To build it, we assume access to two things: (1) a training dataset $\mathcal{D}$ of search trajectories for $N$ planning problems, and (2) a user-specified hybridization factor $x \in [0, 1]$. Given a planning problem of initial state $s_0$, goal state $s_g$, and a gold plan $\mathcal{P}$, let us denote the corresponding gold search trajectory as $\mathcal{T}$, making the training dataset $\mathcal{D} = \{(s_0^{(i)}, s_g^{(i)}), \mathcal{P}^{(i)}, \mathcal{T}^{(i)}\}_{i=1}^N$. In our experiments, the gold plans and the corresponding trajectories will be obtained using a symbolic System-2 Planner (e.g., A* search) but more generally, the source of these trajectories could be any neural or symbolic engine capable of exploration or search. The hybridization factor $x$ is a user-defined hyperparameter that will determine the balance between System-1 and System-2.

## 2.2 System-$1.x$ Overview

System-$1.x$ Planner consists of three trained components: (i) a **Controller** for decomposing planning problems into sub-goals and classifying them as easy or hard, (ii) a **System-1 Planner** for *fast* planning of easy sub-goals, and (iii) a **System-2 Planner** for *slow* planning of hard sub-goals. Fig. 2 shows an overview of our method. We train all three components by automatically generating data from the gold search traces in a way that also respects the user's constraints (specified via the hybridization factor). Below, we describe each component and how they combine to make System-$1.x$.

## 2.3 Training Data Generation for System-$1.x$

**System-1 Planner Training Data.** We denote it as $\mathcal{D}_{\text{Sys1}} = \{(s_0^{(i)}, s_g^{(i)}), \mathcal{P}^{(i)}\}_{i=1}^N$ where the input is a natural language description of the planning problem consisting of a start state $s_0^{(i)}$ and a goal state $s_g^{(i)}$ and the output is the corresponding plan in natural language $\mathcal{P}^{(i)}$ (see Appendix D for examples). For Maze Navigation, these gold plans are also optimal plans (i.e., having the minimum length) while for Blocksworld, they may include additional explorations, making them sub-optimal at times (though always valid).

**System-2 Training Data.** Similar to System-1, the input to a System-2 Planner is also a natural language description of the planning problem. However, the target output of the System-2 Planner is a verbalized search trajectory $\mathcal{T}^{(i)}$ as described in §2.1 (refer to Appendix D for examples). This yields a training dataset of $\mathcal{D}_{\text{Sys2}} = \{(s_0^{(i)}, s_g^{(i)}), \mathcal{T}^{(i)}\}_{i=1}^N$.

**Controller Training Data.** Recall that the System-$1.x$ Planner is designed to minimize state explorations by only performing System-2 planning *as needed*, i.e., saving states with System-1 for easier sub-goals and reserving System-2 for harder sub-goals. Hence, the role of the controller is

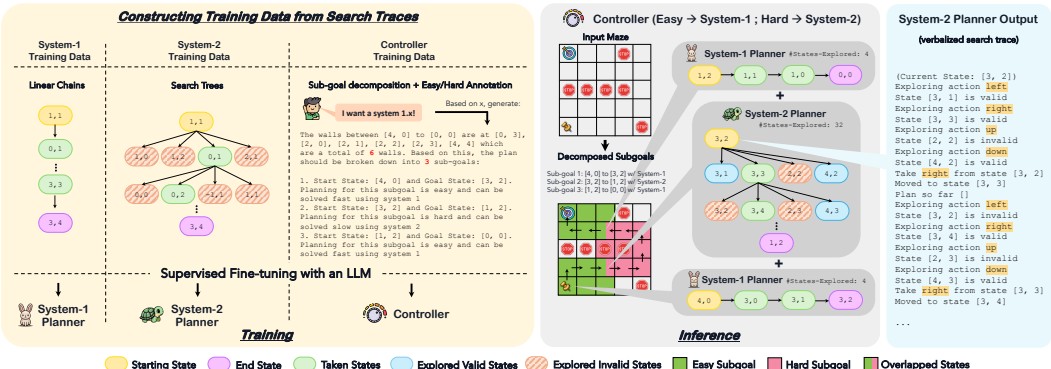

Figure 2: Overview of our System-1.$x$ Planner. **Left:** The process for training data construction. We generate three kinds of data to fine-tune our three components: (1) System-1 data (linear chains), (2) System-2 data (search trees), and (3) controller data (consisting of sub-goals and easy/hard annotations) conditioned on a user-specified hybridization factor $x$. **Right:** During inference, the controller generates the sub-goals and predicts whether each is easy or hard. Then, it invokes System-1 and System-2 for easy and hard sub-goals respectively to construct the final plan.

two-fold: (i) decomposing a problem into sub-goals characterized by a start and end state $(s_i, s_j)$, and (ii) classifying those sub-goals as easy or hard so that System-2 can be invoked only for the hard sub-goals. However, most planning domains lack gold annotations of sub-goal decompositions and their easy and hard classifications. Therefore, we propose an automatic data generation process for training our controller, as described below and more formally in Algorithm 1. The inputs to preparing the controller data are (1) the gold plans $\{\mathcal{P}^{(i)}\}_{i=1}^{N}$ that will be decomposed into sub-goals, (2) the user-defined hybridization factor $x$, and (3) a hardness function $h(\cdot, \cdot)$ for (sub-)goals.

- **Step 1: Defining and Ranking using a Hardness Function.** A hardness function $h(s_i, s_j)$ takes a start state $s_i$ and a goal state $s_j$ as input, and returns a scalar value estimating the hardness of the goal or the sub-goal. An example of a hardness function for a Maze Navigation task is the number of obstacles in a sub-maze, where a higher value indicates greater difficulty. Appendix A and Appendix B more concretely describe and compare these hardness functions for different tasks, respectively. Generally, hardness functions can be constructed for most planning domains where standard heuristics are frequently applied for various search techniques.[4] Using this hardness function between the start and the goal state $h(s_0, s_g)$, we generate a rank ordering of the training examples.

- **Step 2: Annotating Easy Instances as System-1-only.** Intuitively, some planning problems in the train set can be solved successfully via only a System-1 Planner. For such instances, no further decomposition is needed. These correspond to the first $(1 - x) \times N$ easiest samples with the least $h(\cdot, \cdot)$. Note that this is a definition of hardness, and what is considered as easy is determined by the hybridization factor $x$. For example, when $x = 0.0$, all samples will be annotated as easy, making System-1.$x$ the same as System-1.

- **Step 3: Annotating Hard Instances as Hybrid System-1 + System-2.** For the remaining $x \times N$ (hard) samples, we assume that they can be decomposed into sub-goals, each of which can be assigned to either System-1 or System-2. However, to ensure overall task success, the decompositions should neither be too coarse, i.e., requiring further decompositions, nor too fine-grained, making them susceptible to decomposition errors (Prasad et al., 2024). Hence, we adopt a sliding window approach that optimizes for the following: find a contiguous chunk of length $x \times n$ of an $n$-length plan to be assigned System-2, such that it corresponds to the hardest sub-goal and the sub-goals on either side of it correspond to the easiest (and hence assigned System-1). More formally, this requires us to solve the following constrained optimization problem:

$$(j, k) = \operatorname{argmin}_{u,v \in [0,g], u \leq v} h(s_0, s_u) - h(s_u, s_v) + h(s_v, s_g), \text{ s.t. } |v - u| = x \times n$$

Given a start state $s_0$ and goal state $s_g$, we want to find two intermediate states $s_j$ and $s_k$ between $s_0$ and $s_g$ along the length of the plan, such that the hardness of that sub-goal $h(s_j, s_k)$ is maximized (and hence assigned System-2) while the hardness of the other two sub-goals $h(s_0, s_j)$ and $h(s_k, s_g)$

---

[4] https://www.fast-downward.org/Doc/Evaluator

is minimized (and hence assigned System-1). This process results in a three-way decomposition.[5] The final result is a controller training dataset $\mathcal{D}_c = \{(s_0^{(i)}, s_g^{(i)}), \{\mathcal{G}^{(i)}\}_{j=1}^t\}_{i=1}^N$ where for each planning problem, we have an annotated list of $t \in \{1, 2, 3\}$ sub-goals $\{\mathcal{G}_j^{(i)}\}_{j=1}^t$ and their corresponding System-1/System-2 assignments. In practice, the components we use for controller data generation (e.g., sub-goal decomposition strategy, hardness heuristic) can easily be swapped with other variants, based on the problem at hand (further explored in Appendix B).

## 2.4 TRAINING AND INFERENCE OF SYSTEM-1.$x$ PLANNER

Using the data generation process described above, we finetune a single base LLM to behave as a System-1 Planner, a System-2 Planner, or a controller. During inference, we first use the controller to generate a meta-plan – a list of sub-goals and their easy versus hard assignments. After solving each with the appropriate planning model, we concatenate the sub-plans from the two planners, in order, to construct the final plan (see Fig. 2 for an example of a concatenated plan, denoted by '+').

**Train- and Test-time Controllability.** We design System-1.$x$ such that there is a single point of control for training the system, i.e., the hybridization factor $x$, and this control lies with the end-user. Setting it adjusts the training data for the controller, i.e., what is annotated as easy versus hard, both across samples (Step 2 of the algorithm) and within samples (with sub-goals; Step 3 of the algorithm). This ultimately yields the user-specified level of hybridization. Hence, System-1.$x$ offers *training-time control* of compute (e.g., one could train a System-1.25, 1.5, or a 1.75 Planner). Furthermore, the controller's design also offers an additional degree of *test-time control*: since the controller classifies sub-goals as System-1 or 2, the end-user could also re-balance the model towards either system based on the controller confidence threshold. This means that an already trained System-1.5 Planner, for example, can be biased to behave as a System-1.25 or a System-1.75 at test time. In our experiments, we explore how both train-time and test-time control impacts performance.

## 3 EXPERIMENTAL SETUP

**Tasks.** We evaluate System-1.$x$ Planner on two classical planning tasks that are challenging for LLMs (Valmeekam et al., 2023; Lehnert et al., 2024): (1) **Maze Navigation** and (2) **Blocksworld**. In Maze Navigation, given a 2D maze with some cells containing obstacles and four permissible actions {'left', 'right', 'up', 'down'}, the objective is to find a path from a start state to a goal state by avoiding the obstacles. We randomly generate a balanced dataset of 4K planning problems (split into 3200/400/400 samples) with 5x5 mazes, 40% of the cells containing obstacles, and having optimal plan lengths between 1 to 8. Next, to test out-of-distribution (OOD) generalization to longer plans, we experiment with Blocksworld: a task of moving blocks to another goal configuration by only moving one unstacked block at a time. Following the data creation algorithm in Bohnet et al. (2024), we generate problems consisting of 4-7 blocks (without repetition). From there, we create a train/validation/test split of 3000/250/200 samples where the train and the validation split consist of samples with plan lengths 1-6 and the test split consists of samples with plan lengths 7-10.

**Baselines.** We compare our System-1.$x$ Planner with (1) a System-1 Planner, (2) a System-2 Planner, and (3) a System-1.$x$ Planner without sub-goal decomposition. The latter is also a hybrid planner but the controller is trained to predict an instance as either *fully* System-1 or *fully* System-2, thus allowing us to evaluate the effectiveness of sub-goal decomposition of our full System-1.$x$ model. For showing the effectiveness of our neuro-symbolic System-1.$x$ variant, we compare it to A$^*$ by matching their #States-Explored. Under the umbrella of System-2 Planners, we train multiple variants that are fine-tuned with data obtained from different search algorithms. In particular, we consider two uninformed search algorithms (Breadth-First Search and Depth-First Search) and an informed search algorithm (A$^*$). Given the superiority of A$^*$, our primary experiments will be based on A$^*$ traces while we reserve the other search algorithms for further analysis in §4.3. Refer to Appendix A for a short background on A$^*$.

**Evaluation Metrics.** We compare all methods along two axes: plan validity and cost. As defined in §2.1, plan validity is given by the fraction of valid plans generated by a planner while for cost, we compute the average #States-Explored by a planner before reaching the goal. Note that in order

---

[5] Note that when $u = 0$ or $v = g$, there will be two decompositions rather than three.

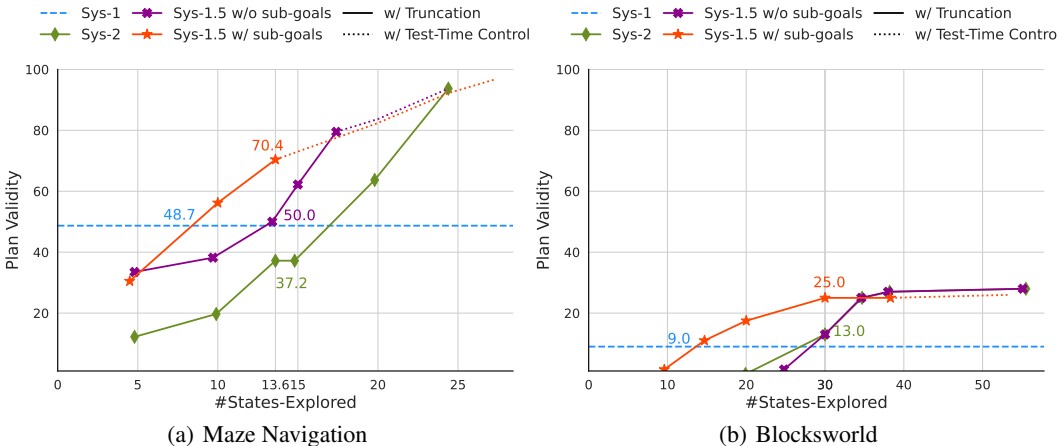

(a) Maze Navigation                    (b) Blocksworld

Figure 3: Comparison of System-1.$x$ Planner (with and without sub-goals) with a System-1 Planner and a System-2 Planner. (a) Maze Navigation: System-1.$x$ with sub-goals matches or generally outperforms all baselines. (b) Blocksworld: For out-of-distribution generalization, System-1.$x$ with sub-goals leads to much higher plan validity at lower #States-Explored.

to fairly compare all systems, their #States-Explored should be matched, such that we compare performance at a fixed budget for each system. First, since System-1 does not do any exploration, it is constant w.r.t. budget and hence has the same plan validity irrespective of #States-Explored. Next, to adapt System-2 and System-1.$x$ to lower budgets, we truncate the search when the maximum number of states allowed by the budget has been reached. This value of maximum allowed states is given by the highest possible value such that the average #States-Explored matches the desired budget. We refer to this as **truncation** below. Finally, to adapt System-1.$x$ to higher budgets, we use System-1.$x$'s test-time controllability, increasing the bias term on System-2 reasoning (cf. §2.4). As this term increases, the model uses more and more System-2 planning, i.e., $x \to 1$ in System-1.$x$. We refer to this as **test-time control**. We perform this matching and report plan validity for the following values of #States-Explored. First, we vary the maximum budget of states explored in intervals of 5 for Maze and 10 for Blocksworld, up to the number of states used by System-2 (the most compute-intensive system we consider). We also consider two specific points in the range of #States-Explored: we match System-2's #States-Explored to the number of states that System-1.$x$ uses by default. In the other direction, we match System-1.$x$'s #States-Explored to the number of states used by System-2. These points let us directly compare the two systems without any truncation or test-time control. In the main paper, we present our results with plots while we refer to the Appendix E for detailed tables.

**Implementation Details.** We choose Mistral-7B-Instruct-v0.2 (Jiang et al., 2023) as the base LLM and fine-tune all our components with LoRA (Hu et al., 2021) with a rank of 8 for a maximum of 3 epochs and a batch size of 4, resulting in three adapters for System-1, System-2, and the controller. For the purposes of §4.1, System-1.$x$ will specifically refer to a System-1.5 Planner (i.e., $x = 0.5$, which is a balanced hybrid between a System-1 Planner and System-2 Planner) and fine-tuned from A* traces. Refer to Appendix A for additional implementation details e.g., our hardness functions and System-2 implementations.

## 4 RESULTS AND ANALYSIS

### 4.1 SYSTEM-1.$x$ OUTPERFORMS SYSTEM-1 AND SYSTEM-2

**Study Setup.** In this first set of experiments, we compare our System-1.$x$ Planner with (1) a System-1 Planner, (2) a System-2 Planner, and (3) a System-1.$x$ Planner without sub-goal decomposition.

**Results for Maze Navigation.** Fig. 3(a) (cf. Table 3) shows the plan validity obtained by each planner at different #States-Explored. The values of #States-Explored are as described previously in §3. Based on this, we summarize our key findings.

- **System-1.$x$ outperforms all baselines at all budgets.** Without truncation or test-time control, System-1.$x$ ($x = 0.5$) uses 13.6 states on average. At this number of states, it generates 70.4%

valid plans, outperforming the System-1 Planner by 21% (48.7% → 70.4%), the System-2 Planner by 33% (37.2% → 70.4%), and the System-1.$x$ variant without sub-goal decomposition by 20% (50.0% → 70.4%). Next, to compare all methods at *lower* budgets (i.e., with #States-Explored of 5 and 10), we do state truncation (as described in §3) by limiting the search (generation) to a maximum number of states. We find that even at these lower budgets, System-1.$x$ continues to be significantly more accurate compared to baselines, pointing to an effective allocation of resources. Finally, System-1.$x$'s *test-time control* capability lets us increase compute to match that of System-2 by biasing the controller and solving more sub-goals with System-2. Doing so eventually transforms a System-1.$x$ Planner into a *System-2 Planner with sub-goal decomposition*, outperforming vanilla System-2 by 3% and achieving the highest validity of 96.7% at 27.3 states. This highlights the effectiveness of our sub-goal decomposition (more details below).

- **System-1.$x$ benefits from sub-goal decomposition.** We validate the utility of sub-goal decomposition from two observations. First, the full System-1.$x$ outperforms the System-1.$x$ variant that does not perform sub-goal decomposition. Second, System-2 also benefits from sub-goal decomposition as shown by the maximum validity of 96.7% which is achieved via *test-time control* and biasing the controller to solve every sub-goal using System-2. However, note that System-2 with decomposition still applies System-2 planning to every step, making System-1.$x$ more performant at any given budget.

- **System-1 is cheaper but not accurate.** On one extreme of the plot, System-1 Planner is cheaper since it only explores the states that are part of its plan, amounting to an average of 3 states. However, this comes at a cost to performance: it also only generates 49% valid plans. Another limitation of a System-1 Planner is that unlike System-1.$x$, it is not controllable and hence, there is no easy affordance for improving performance by exploring more states.

- **System-2 is more accurate but expensive.** On the other extreme, System-2 Planner achieves a maximum validity of 94% but explores a much higher 24.4 states in the process. While System-2 can be adapted to minimize #States-Explored, such adaptivity is only possible during inference via state-truncation. Test-time truncation of #States-Explored is typically ad-hoc and not preferable because the loss in performance will only increase as the plan length grows. This is in contrast to the System-1.$x$ Planner, which allows for systematic control of compute both at training-time (via a user-defined hybridization factor $x$), and at test-time (by biasing the controller).

**Results for Blocksworld.** Fig. 3(b) (cf. Table 4) reports our results on Blocksworld (BW) that studies out-of-distribution generalization to longer plan lengths. Our first observation is that System-1 only obtains a plan validity of 9% and although System-2's validity is higher at 28%, given the longer plan lengths, it also explores a significantly higher number of states (55). Thus, when we perform truncation to get lower values of #States-Explored, System-2's valid plans drops to almost 0%. System-1.$x$ without sub-goal decomposition also does not work for OOD test points because the controller predicts almost all samples as hard and solves them using System-2. Hence, System-1.$x$ without sub-goal decomposition practically becomes a System-2 in such scenarios, suffering major losses to performance with truncation. However, in contrast to all baselines, System-1.$x$ *with* sub-goal decomposition can solve a larger fraction of problems at lower budgets (e.g., 25% at 30 states compared to System-2 Planner's 13%) because of its ability to leverage a System-1 Planner for simpler sub-goals that have shorter plan lengths and are more likely to be in-distribution (refer to example in Fig. 9). At maximum #States-Explored, System-1.$x$ obtains comparable performance to System-2. Overall, when testing on OOD data, the results in Fig. 3(b) highlight the key role that the System-1.$x$ controller plays in decomposing longer-horizon planning problems. Crucially, what is generally a weak System-1 Planner for harder, OOD goals can be used effectively to solve easier, ID sub-goals, thereby saving a lot of compute.

## 4.2 NEURO-SYMBOLIC SYSTEM-1.$x$ OUTPERFORMS OR MATCHES A* AT ALL STATE BUDGETS

**Study Setup.** While we primarily designed System-1.$x$ as a fully neural planner with all three components (System-1, 2, and the controller) developed on top of the same LLM, we note that System-1.$x$ can also act as a neuro-symbolic planner. This can be achieved by using a symbolic solver like A* as the System-2 component. We follow this setup to report results on the Maze Navigation task, again setting $x = 0.5$.

**Results.** Fig. 4 (cf. Table 5) compares the validity of all planners at different #States-Explored. We note that an LLM-based planner may not be directly comparable to symbolic

planners like A$^*$ with respect to our #States-Explored metric because the former generates tokens for states which computationally can be more expensive than invoking a symbolic planner. However, for the purpose of this study, we compare them *only based on the number of explored states, without considering the underlying computation responsible for such explorations*. Our main result is that the conclusions drawn with a fully neural System-1.$x$ also transfer to a neuro-symbolic System-1.$x$. Notably, at an average of 11.6 states, neuro-symbolic System-1.$x$ outperforms A$^*$ by a large margin of 39% (31.0% $\rightarrow$ 70.5%). This can again be explained by the ineffectiveness of #States-Explored truncation for System-2 Planners, symbolic or neural. The trend is also consistent at all lower budgets. System-1.$x$ can be test-time controlled to obtain a near-perfect validity (99.2%), matching that of A$^*$.

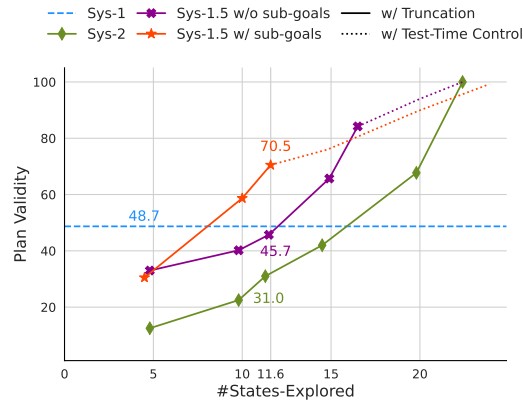

Figure 4: Performance of System-1.$x$ in conjunction with a symbolic (A$^*$) System-2 Planner on Maze.

### 4.3 ANALYSES AND ABLATIONS OF SYSTEM-1.$x$ PLANNER

We conduct all analyses and ablations on the maze navigation task.

**Train-time Control: System-1.75 trades off efficiency for plan validity compared to System-1.5.** One of the core strengths of our planner is its *training-time controllability*. The hybridization factor $x$ allows the user to specify the System-1 to System-2 balance they want in the final planner. To demonstrate this capability, we train a System-1.75 Planner by setting $x = 0.75$. Table 6 (cf. Table 6) shows our results on the maze task. By default, System-1.75 uses 16.6 states on average and generates 75.7% valid plans, compared to System-1.5's 70.4% with a default average of 13.6 states (Table 3) i.e., a 5% gain at the cost of 3 more states.

Thus, as we increase the value of $x$, the System-1.$x$ Planner will use more states and improve performance, converging to a System-2 Planner that additionally performs sub-goal decomposition. Compared to System-2, System-1.75 is again significantly more performant at all budgets (e.g., by 28% at 16.6 states) and eventually obtains the highest fraction of valid plans at 96.7% when acting as a System-2 with sub-goals. This echoes our System-1.5 Planner findings, which also demonstrated similar gains at high budgets.

Broadly, these results showcase our controller's effectiveness, which in turn validates our hardness measures and sub-goal decompositions. We further ablate features of the controller in Appendix B, comparing against a random controller, testing the effectiveness of the sliding window, and contrasting different hardness functions.

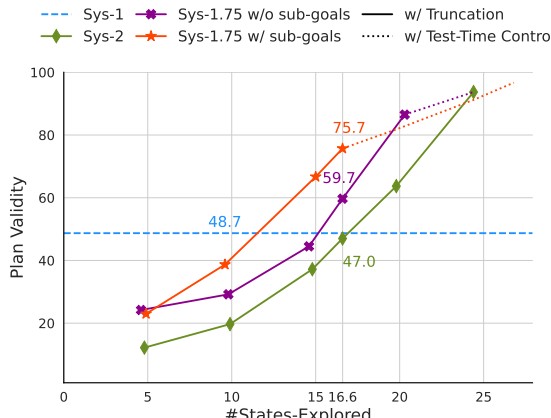

Figure 5: Train-time controllability of System-1.$x$ Planner by setting $x = 0.75$. System-1.75 largely outperforms System-2. Compared to Fig. 3(a), System-1.75 yields 75.7% valid plans vs. 70.4% (with System-1.5) by utilizing 3 more states.

**Generalizability: System-1.$x$ beats System-2 with all search algorithms (A\*, DFS, BFS).** We conducted our main experiments by leveraging supervision from A$^*$ traces. In Appendix Fig. 6, we show that System-1.$x$ is robust to different search algorithms, working with other options like Breadth First Search (BFS) and Depth First Search (DFS) and outperforming baselines by large margins (see Appendix B for details). In Appendix C, we show a representative example of System-1.$x$ outperforming the baseline systems.

## 5  RELATED WORK

**Combining LLMs and Symbolic Planners.** A large body of recent work has highlighted the shortcomings of LLMs on long-horizon planning problems (Valmeekam et al., 2023; 2024; Pallagani et al., 2023; Momennejad et al., 2024; Hirsch et al., 2024; Zheng et al., 2024; Aghzal et al., 2023), persisting across popular prompting techniques like Chain-of-Thought (Wei et al., 2022), ReAct (Yao et al., 2022), and Reflexion (Shinn et al., 2024). These methods largely resemble System-1 inference-time methods in which the LLM does not explicitly search, backtrack, or learn from incorrect actions. Systematic search is either not performed at all or delegated to symbolic solvers (e.g., Liu et al., 2023; Pan et al., 2023; Xie et al., 2023, discussed further below), thus deviating from the focus of this work on equipping LLMs to solve harder planning problems via search. More related to our work are inference-time methods that *do* involve planning, tree search, and backtracking. These System-2-inspired methods have a search algorithm that operates *on top of* and guides an LLM with reward models, value functions, or verifiers (Hao et al., 2023; Yao et al., 2024; Besta et al., 2024; Zhou et al., 2023; Sel et al., 2023; Koh et al., 2024). In a similar spirit, neuro-symbolic planning systems like *LLM-Modulo Frameworks* (Kambhampati et al., 2024) have emerged that combine LLMs with symbolic planners (Nye et al., 2021; Liu et al., 2023; Pan et al., 2023; Xie et al., 2023; Zuo et al., 2024; Fabiano et al., 2024; Katz et al., 2025; Zhuang et al., 2024). System-1.$x$ differs from this class of inference-time methods in two major ways. Firstly, it is a training-time method that teaches the LLM to search, with the goal of enhancing the LLM's intrinsic planning capabilities and discovering more effective search strategies, making it self-contained (as opposed to complex inference-time methods relying on external modules). Secondly, it is also a hybrid system that can alternate between System-1 and System-2-style planning, whereas past work is capable of one or the other, but not both. Notably, this kind of hybridization distinguishes System-1.$x$ from other work that either creates hybrids between weaker and stronger models in the form of model cascades (Lin et al., 2024; Yue et al., 2023) or between neural and symbolic methods (e.g., LLM-Modulo frameworks).

**LLMs Generating Plans.** Methods combining LLMs and symbolic planners need some communication channel between the LLM and the symbolic planning component. This is usually a value function or reward model producing a single score for a state, making the method's success contingent on the quality of the reward model, or the LLM's ability to act as a value function (as in Yao et al. (2024); Lightman et al. (2023)). Furthermore, off-loading planning or search adds additional modules at test-time that may not always be available for certain domains. It also assumes that the model's representation is already sufficient for the task and complicates finetuning the representation, since externalized search is typically non-differentiable. In an attempt to mitigate such concerns, a second line of work removes the dependency on separate reward models and external tools and teaches LLMs to conduct search by themselves (Yang et al., 2022; Gandhi et al., 2024; Lehnert et al., 2024). For example, Gandhi et al. (2024) introduce Stream-of-Search (SoS), which teaches LLMs to search by verbalizing trajectories and Lehnert et al. (2024) propose Searchformer, that learns from A$^*$ search dynamics. This line of work predates LLMs: Anthony et al. (2017) generate data from tree search which they use to iteratively train a neural model to perform System-1 planning. However, Gandhi et al. (2024) and Lehnert et al. (2024) differ from our work in that their final models are System-2-only, incurring a high computational cost for all examples and states by performing full planning at all times, whereas our method is a hybrid between Systems 1 and 2, dynamically allocating computation to more difficult parts of the plan. Moreover, to meet user budget constraints, System-2-only methods also have to rely on approaches like post-hoc truncation while System-1.$x$ offers more systematic controllability via its controller.

## 6  CONCLUSION

We introduced the System-1.$x$ Planner, a novel LLM-based planner that adaptively interleaves quick and cost-efficient System-1 planning with more costly but performant System-2 planning. On two tasks, we showed that our method results in strong performance across a range of budgets: on Mazes, System-1.$x$ obtains the best performance at every budget, and on Blocksworld it outperforms System-2 at all budgets except the highest, where it is comparable. In our analysis, we find that System-1.$x$ is controllable, generalizes to various search algorithms, and performs better out-of-distribution.

ETHICS STATEMENT

Large Language Models have been shown to reflect stereotypes, biases, and other negative traits present in their pre-training data (Weidinger et al., 2021). Hence, the outputs produced by our fine-tuned planner may exhibit undesirable behavior similar to the base model and have the same potential for misuse as other fine-tuned LLMs. Hence, more studies are needed to evaluate and mitigate such biases in LLMs.

REPRODUCIBILITY STATEMENT

We are making our code and data available in the supplementary material to enable replication of our findings. We also show the exact prompts in Appendix D.

ACKNOWLEDGMENTS

This work was supported by NSF-CAREER Award 1846185, NSF-AI Engage Institute DRL-2112635, DARPA MCS Grant N66001-19-2-4031, and Google PhD Fellowships. The views contained in this article are those of the authors and not of the funding agency.

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

# A    ADDITIONAL DETAILS OF SYSTEM-1.$x$ PLANNER

---

**Algorithm 1** Training Data Generation for System-1.$x$ Controller

---

**Input:** System-1 Training Data $\mathcal{D}_{\text{Sys1}} = \{(s_0^{(i)}, s_g^{(i)}), \mathcal{P}^{(i)}\}_{i=1}^N$, Hybridization Factor $x \in [0,1]$, Hardness Function $h(\cdot, \cdot)$
**Output:** Training Data $\mathcal{D}_c$ for System-1.$x$ Controller
$\texttt{sort}(\mathcal{D}_{\text{Sys1}}, \texttt{key}=h(s_0, s_g))$ // *Ranking data points in increasing order of hardness*
$\mathcal{D}_c \leftarrow \{\}$
**for** data point $((s_0^{(i)}, s_g^{(i)}), \mathcal{P}^{(i)}) \in \mathcal{D}_{\text{Sys1}}$ **do**
  **if** $i < (1-x) \times N$ **then**
    *// For the first $(1-x)\%$ of instances (easiest), solve them directly using System-1 planner*
    $y^{(i)} \leftarrow \{(s_0^{(i)}, s_g^{(i)}), \text{``Sys1''}\}$
  **else**:
    *// For the remaining $x\%$ hardest instances, find the best decomposition into three sub-goals such that*
    *the middle sub-goal $(s_j, s_k)$ is solved using System-2 and has a length of $x\%$ of the total plan length*
    $(j, k) = \text{argmin}_{u,v \in [0,g], u \leq v}(h(s_0^{(i)}, s_u) - h(s_u, s_v) + h(s_v, s_g^{(i)}))$ s.t. $|v - u| = x \times n$
    $y^{(i)} \leftarrow \{(s_0^{(i)}, s_j), \text{``Sys1''}\}$
    $y^{(i)} \leftarrow y^{(i)} + \{(s_j, s_k), \text{``Sys2''}\}$
    $y^{(i)} \leftarrow y^{(i)} + \{(s_k, s_g^{(i)}), \text{``Sys1''}\}$
  $\mathcal{D}_c \leftarrow \mathcal{D}_c + \{(s_0^{(i)}, s_g^{(i)}), y^{(i)}\}$
**return** $\mathcal{D}_c$

---

## A.1    BACKGROUND ON A*

A* is an informed search algorithm that decides the next best state to expand with a function $f(n) = g(n) + t(n)$ that considers (1) the cost $g(n)$ from the start state to the current state $n$, and (2) a heuristic function $t(n)$ that gives an estimate of the cost from state $n$ to the goal. Note that A* is optimal if the heuristic $t(n)$ is admissible (i.e., $t(n) \leq t^*(n)$ which means that the heuristic never overestimates the true cost to the goal $h^*(n)$) and consistent (i.e., $t(n) \leq c(n, n') + t(n')$ for all states $n$ and its successor $n'$ and $c$ is the cost to transition from $n$ to $n'$). For the Maze Navigation task, Manhattan Distance is an example of such an admissible heuristic, and hence A* with it will always generate optimal plans.

## A.2    IMPLEMENTATION DETAILS FOR MAZE NAVIGATION

**System-2 Planner.**  We implement a System-2 Planner for the Maze domain by verbalizing all possible explorations at each state, including the ones that lead to invalid states (i.e., outside the maze, have obstacles, or have already been visited). See example in Appendix D.

**Hardness Function.**  Given two maze states $s_i = (a, b)$ and $s_j = (c, d)$ we define hardness of a sub-goal $h(s_i, s_j)$ as the number of obstacles in the rectangular sub-maze given the end-points. More obstacles will mean that the model might have to search more and would likely benefit from System-2 planning.

## A.3    IMPLEMENTATION DETAILS FOR BLOCKSWORLD

**System-2 Planner.**  In contrast to the Maze domain, the number of actions at each state in Blocksworld problem is directly proportional to the total number of blocks. Consequently, the number of possible states that can be explored (both valid and invalid) is significantly larger in this domain, often exceeding the maximum input token budgets of standard LLMs (in our case, 8096 tokens). Therefore, when verbalizing the search trajectories $\mathcal{T}$ in $\mathcal{D}_{\text{Sys2}}$ using A* search to train System-2 Planner, we cap the number of valid and invalid states explored to 3 and 2 respectively. The verbalized valid states are chosen by picking the top-3 scored states as per the A* heuristic (that computes the number of mismatches between the start state and the goal state), while (up to) 2 invalid

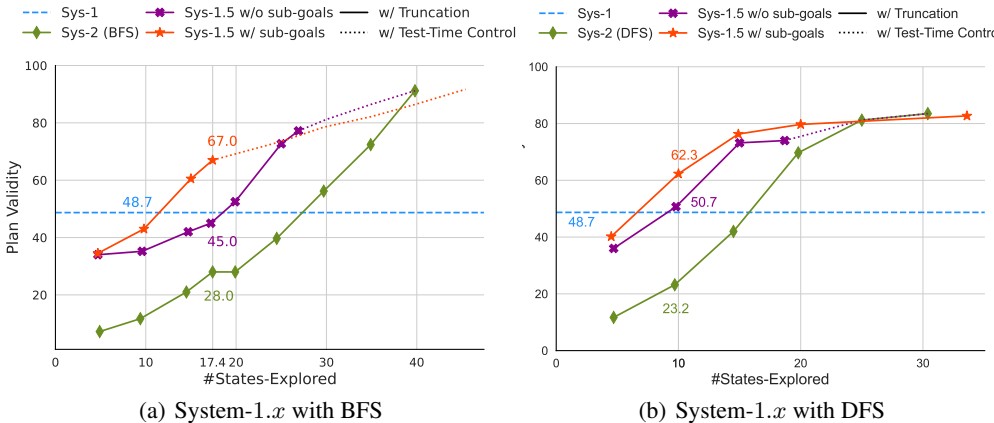

(a) System-1.$x$ with BFS  (b) System-1.$x$ with DFS

Figure 6: Comparison of System-1.$x$ Planner on Maze Navigation using (a) Breadth-First Search and (b) Depth-First Search traces. The System-1.$x$ Planner generally outperforms System-2 Planner in both cases, converging to System-2, showing robustness to the underlying search algorithm.

states are chosen randomly to make the model learn from mistakes. Refer to Appendix D for an example.

**Hardness Function.** A state in Blocksworld is characterized by the configuration of the blocks. Our hardness function computes a distance metric for a sub-goal: For every block that is not in its correct position in the goal state (i.e., the blocks above and below it are different between the start and goal states), we add a cost of 1. Additionally, for a block that is *not* on the table and also not in its correct position, we add 1 more to the cost, indicating greater hardness for moving blocks that are in the middle of a stack.

## B  ADDITIONAL RESULTS

**Generalizability: System-1.$x$ beats System-2 with all search algorithms (A*, DFS, BFS).** We conducted our main experiments by leveraging supervision from A* traces. In this experiment, we show that System-1.$x$ is robust to different search algorithms, working with other options like Breadth First Search (BFS) and Depth First Search (DFS). First, note that for mazes, BFS explores all four actions (up, down, left, right) at each state and hence, on average, is the most expensive search strategy in terms of states explored. However, for an unweighted maze, BFS always generates optimal plans. DFS, on the other hand, explores fewer states because it returns the first valid plan, though that plan may not necessarily be optimal. Appendix D contains examples of BFS and DFS trajectories. Based on Fig. 6 (cf. Tables 7 and 8), we summarize our key findings:

- System-1.$x$ continues to outperform System-2 and 'System-1.$x$ without sub-goal decomposition' by large margins (e.g., up to 39% at 17.4 states for BFS and at 10 states for DFS), showcasing our planner's flexibility to learn from different algorithms.
- System-1.$x$ closely resembles the behavior of the search algorithm it is trained on. For example, System-1.$x$ (BFS) is more accurate and generates a significantly higher number of optimal plans than System-1.$x$ (DFS) because the former is trained on optimal plans (refer to Table 2 for a study on validity versus optimality). In the process, however, System-1.5 (BFS) explores more states than System-1.$x$ (DFS), similar to their symbolic counterparts. Among the three algorithms studied in this paper, System-1.$x$ (A*) is the most accurate and efficient.

**System-1.$x$ Controller outperforms a random controller.** In Table 1, we compare our System 1.5 controller with a random controller that arbitrarily chooses 50% problems to be solved by System-2. Our trained controller outperforms the random controller by a significant 10%.

**Analyses of System-1.$x$ Controller.** In this section, we analyze the different components of our System-1.$x$ controller.

Table 1: Comparison of our trained controller in System-1.$x$ Planner with a controller that randomly chooses $x\%$ of problems to solve with System-2.

|  | Plan Validity |
|---|---|
| System 1.5 (w/ random controller) | 70.5 |
| System 1.5 (w/ our controller) | 79.5 |

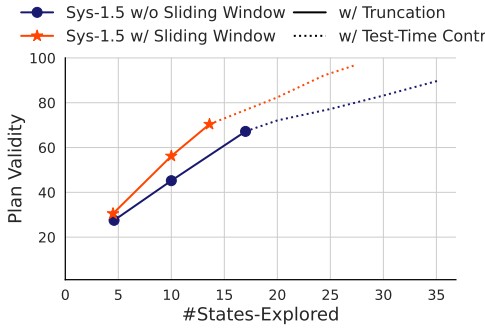

Figure 7: Comparison of System-1.$x$ Planner with and without sliding window-based sub-goal decomposition on the Maze Navigation task. Sliding window-based decomposition outperforms the variant without it at all budgets.

Figure 8: Comparison of two hardness functions (Manhattan Distance and #Obstacles) for Maze Navigation. Both yield comparable results, with #Obstacles being slightly better.

- **Effectiveness of Sliding Window Decomposition.** Recall that we generated training data for the controller by using a sliding window approach that assigned a contiguous $x\%$ of the plan to System-2 according to a hardness function. Here, we evaluate the effectiveness of this approach by comparing it to a System-1.$x$ variant *without* the sliding window decomposition. In this variant, the $x\%$ System-2 allocation can either be at the beginning or at the end of the plan but not in the middle, resulting in a maximum of two decompositions instead of three. As shown in Fig. 9, our sliding window variant not only obtains higher plan validity but is also far more efficient. This is because with three decompositions, a weaker System-1 Planner can be used to solve two granular sub-goals (instead of one) and hence is more likely to succeed.
- **Comparison of Hardness Functions.** Our experiments on maze used the number of obstacles in a sub-maze as the hardness function. In Table 10, we compare it to Manhattan Distance, which rates goals that are further away (according to the distance function) as more difficult. We observe that both functions lead to comparable performance, with obstacles being slightly more efficient. Taken together with the previous experiment, these results quantify the influence of two essential inputs to the controller's training: hardness function and the decomposition strategy.

Table 2: Comparison of Plan Validity and Optimality for System-1.$x$ Planner in Maze Navigation using A$^*$ search. While not using sub-goal decomposition nearly guarantees optimality, using sub-goal decomposition may generate plans that are sub-optimal.

|  | Plan Validity | Plan Optimality | #States-Explored |
|---|---|---|---|
| System-1.$x$ (w/o sub-goals) | 33.5 | 33.2 | 4.8 |
| System-1.$x$ (w/o sub-goals) | 38.2 | 38.0 | 9.7 |
| System-1.$x$ (w/o sub-goals) | 62.2 | 61.7 | 15.0 |
| System-1.$x$ (w/o sub-goals) | 79.5 | 78.7 | 17.4 |
| System-1.$x$ (w/ sub-goals) | 30.5 | 30.2 | 4.5 |
| System-1.$x$ (w/ sub-goals) | 56.2 | 53.7 | 10.0 |
| System-1.$x$ (w/ sub-goals) | 70.4 | 61.5 | 13.6 |

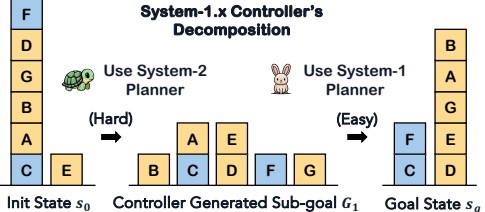

Figure 9: An example decomposition generated by System 1.5 for Blocksworld. The controller generates an intermediate state $\mathcal{G}_1$ where blocks 'C' and 'F' are a block apart, with the sub-goal $s_0 \rightarrow \mathcal{G}_1$ solved using System-2 and $\mathcal{G}_1 \rightarrow s_g$ solved via System-1.

**Optimality of System-1.$x$.** In Table 2, we compare the optimality and validity of the plans generated by our System-1.$x$ Planner. While System-1.$x$ without sub-goal decomposition generates plans that are almost always optimal, we observe that sub-goal decomposition may sometimes hurt optimality. This happens because the sub-goals generated by the controller may not always lie on the optimal path.

## C   QUALITATIVE ANALYSIS OF SYSTEM-1, SYSTEM-2, AND SYSTEM-1.$x$

Fig. 10 shows an example from our test set where both System-1 and System-2 fail but System-1.$x$ succeeds. We consider a maze configuration where the plan length is quite long (8) and there is a solitary path to the goal, making the problem challenging for LLMs. We find that the System-1 Planner generates an incorrect plan that ignores the placement of the obstacles. On the other hand, the System-2 Planner avoids obstacles successfully initially but then takes a path that leads it to a dead end. It is unable to recover from that and the search does not terminate. Our System-1.$x$ Planner is, however, able to solve the problem successfully by first breaking it up into 3 sub-goals and solving the easy sub-goals close to the start state and the goal state using System-1 and the hard sub-goal in the middle (that includes a number of obstacles) with deliberate System-2 planning.

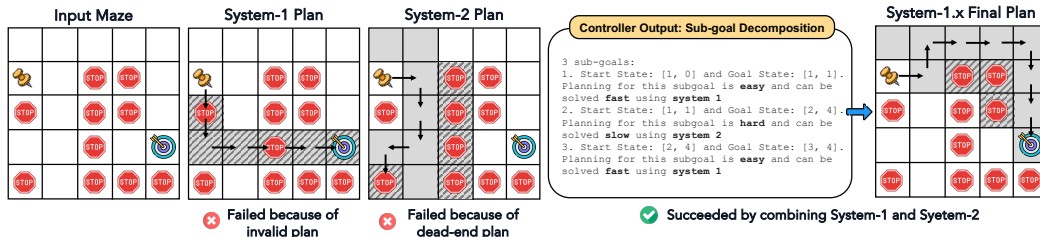

Figure 10: A qualitative example showing (1) a failure case for a System-1 Planner because it generates a plan through the obstacles, (2) a failure case for a System-2 Planner because the generated plan leads to a dead end and the model does not recover from it, and (3) a successful case for our System-1.$x$ Planner that solves the easy sub-goals near the start state and the goal state using System-1 and the hard sub-goal in the middle using System-2.

## D   PROMPTS AND EXAMPLES

See the subsequent pages for prompts and examples from both domains.

**System-1 / System-2 Prompt for Maze**

You are in a 2d maze of dimensions **5** and **5** and some of the cells have walls. The walls are placed in cells [[3, 4], [0, 4], [3, 3], [1, 1], [2, 3], [0, 0], [2, 0], [4, 0],[1, 0], [2, 4]].

Given a start and a goal state, your task is to **generate the optimal plan as a sequence of actions.** The optimal plan is one that has the minimum number of steps. The list of permissible actions that you can take at any given cell are ['left', 'right', 'up', 'down'].

The optimal plan from **[1, 4]** to **[3, 2]** is

**System-1 Output for Maze (#States-Explored=4)**

start [1, 4] | left [1, 3] | left [1, 2] | down [2, 2] | down [3, 2]

**System-2 (A*) Output for Maze (#States-Explored=16)**

Moved to state [1, 4] | Plan so far [] | Exploring action 'left' to move to state [1, 3] | State [1, 3] is valid | Exploring action 'right' to move to state [1, 5] | State [1, 5] is invalid | Exploring action 'up' to move to state [0, 4] | State [0, 4] is invalid | Exploring action 'down' to move to state [2, 4] | State [2, 4] is invalid | Taking action 'left' from state [1, 4] | Moved to state [1, 3] | Plan so far ['left'] | Exploring action 'left' to move to state [1, 2] | State [1, 2] is valid | Exploring action 'right' to move to state [1, 4] | State [1, 4] is invalid | Exploring action 'up' to move to state [0, 3] | State [0, 3] is valid | Exploring action 'down' to move to state [2, 3] | State [2, 3] is invalid | Taking action 'left' from state [1, 3] | Moved to state [1, 2] | Plan so far ['left', 'left'] | Exploring action 'left' to move to state [1, 1] | State [1, 1] is invalid | Exploring action 'right' to move to state [1, 3] | State [1, 3] is invalid | Exploring action 'up' to move to state [0, 2] | State [0, 2] is valid | Exploring action 'down' to move to state [2, 2] | State [2, 2] is valid | Taking action 'down' from state [1, 2] | Moved to state [2, 2] | Plan so far ['left', 'left', 'down'] | Exploring action 'left' to move to state [2, 1] | State [2, 1] is valid | Exploring action 'right' to move to state [2, 3] | State [2, 3] is invalid | Exploring action 'up' to move to state [1, 2] | State [1, 2] is invalid | Exploring action 'down' to move to state [3, 2] | State [3, 2] is valid | Taking action 'down' from state [2, 2] | Moved to state [3, 2] | Plan so far ['left', 'left', 'down', 'down'] | **Goal state [3, 2] reached!**

**System-2 Output (BFS) for Maze (#States-Explored=36)**

```
Moved to state [1, 4] | Plan so far [] | Exploring action 'left' to move to
state [1, 3] | State [1, 3] is valid | Exploring action 'right' to move to
state [1, 5] | State [1, 5] is invalid | Exploring action 'up' to move to
state [0, 4] | State [0, 4] is invalid | Exploring action 'down' to move to
state [2, 4] | State [2, 4] is invalid | Taking action 'left' from state [1,
4] | Moved to state [1, 3] | Plan so far ['left'] | Exploring action 'left' to
move to state [1, 2] | State [1, 2] is valid | Exploring action 'right' to
move to state [1, 4] | State [1, 4] is invalid | Exploring action 'up' to move
to state [0, 3] | State [0, 3] is valid | Exploring action 'down' to move to
state [2, 3] | State [2, 3] is invalid | Taking action 'left' from state [1,
3] | Moved to state [1, 2] | Plan so far ['left', 'left'] | Exploring action
'left' to move to state [1, 1] | State [1, 1] is invalid | Exploring action
'right' to move to state [1, 3] | State [1, 3] is invalid | Exploring action
'up' to move to state [0, 2] | State [0, 2] is valid | Exploring action 'down'
to move to state [2, 2] | State [2, 2] is valid | Taking action 'up' from
state [1, 3] | Moved to state [0, 3] | Plan so far ['left', 'up'] | Exploring
action 'left' to move to state [0, 2] | State [0, 2] is valid | Exploring
action 'right' to move to state [0, 4] | State [0, 4] is invalid | Exploring
action 'up' to move to state [-1, 3] | State [-1, 3] is invalid | Exploring
action 'down' to move to state [1, 3] | State [1, 3] is invalid | Taking
action 'up' from state [1, 2] | Moved to state [0, 2] | Plan so far ['left',
'left', 'up'] | Exploring action 'left' to move to state [0, 1] | State [0, 1]
is valid | Exploring action 'right' to move to state [0, 3] | State [0, 3] is
invalid | Exploring action 'up' to move to state [-1, 2] | State [-1, 2] is
invalid | Exploring action 'down' to move to state [1, 2] | State [1, 2] is
invalid | Taking action 'down' from state [1, 2] | Moved to state [2, 2] |
Plan so far ['left', 'left', 'down'] | Exploring action 'left' to move to
state [2, 1] | State [2, 1] is valid | Exploring action 'right' to move to
state [2, 3] | State [2, 3] is invalid | Exploring action 'up' to move to
state [1, 2] | State [1, 2] is invalid | Exploring action 'down' to move to
state [3, 2] | State [3, 2] is valid | Taking action 'left' from state [0, 3]
| Moved to state [0, 2] | Plan so far ['left', 'up', 'left'] | Exploring
action 'left' to move to state [0, 1] | State [0, 1] is valid | Exploring
action 'right' to move to state [0, 3] | State [0, 3] is invalid | Exploring
action 'up' to move to state [-1, 2] | State [-1, 2] is invalid | Exploring
action 'down' to move to state [1, 2] | State [1, 2] is invalid | Taking
action 'left' from state [0, 2] | Moved to state [0, 1] | Plan so far ['left',
'left', 'up', 'left'] | Exploring action 'left' to move to state [0, 0] |
State [0, 0] is invalid | Exploring action 'right' to move to state [0, 2] |
State [0, 2] is invalid | Exploring action 'up' to move to state [-1, 1] |
State [-1, 1] is invalid | Exploring action 'down' to move to state [1, 1] |
State [1, 1] is invalid | Taking action 'left' from state [2, 2] | Moved to
state [2, 1] | Plan so far ['left', 'left', 'down', 'left'] | Exploring action
'left' to move to state [2, 0] | State [2, 0] is invalid | Exploring action
'right' to move to state [2, 2] | State [2, 2] is invalid | Exploring action
'up' to move to state [1, 1] | State [1, 1] is invalid | Exploring action
'down' to move to state [3, 1] | State [3, 1] is valid | Taking action 'down'
from state [2, 2] | Moved to state [3, 2] | Plan so far ['left', 'left',
'down', 'down'] | Goal state [3, 2] reached!
```

**System-2 Output (DFS) for Maze (#States-Explored=29)**

```
Moved to state [1, 4] | Plan so far [] | Exploring action 'left' to move to
state [1, 3] | State [1, 3] is valid | Taking action 'left' from state [1, 4]
| Moved to state [1, 3] | Plan so far ['left'] | Exploring action 'left' to
move to state [1, 2] | State [1, 2] is valid | Taking action 'left' from state
[1, 3] | Moved to state [1, 2] | Plan so far ['left', 'left'] | Exploring
action 'left' to move to state [1, 1] | State [1, 1] is invalid | Exploring
action 'right' to move to state [1, 3] | State [1, 3] is invalid | Exploring
action 'up' to move to state [0, 2] | State [0, 2] is valid | Taking action
'up' from state [1, 2] | Moved to state [0, 2] | Plan so far ['left', 'left',
'up'] | Exploring action 'left' to move to state [0, 1] | State [0, 1] is
valid | Taking action 'left' from state [0, 2] | Moved to state [0, 1] | Plan
so far ['left', 'left', 'up', 'left'] | Exploring action 'left' to move to
state [0, 0] | State [0, 0] is invalid | Exploring action 'right' to move to
state [0, 2] | State [0, 2] is invalid | Exploring action 'up' to move to
state [-1, 1] | State [-1, 1] is invalid | Exploring action 'down' to move to
state [1, 1] | State [1, 1] is invalid | Exploring action 'right' to move to
state [0, 3] | State [0, 3] is valid | Taking action 'right' from state [0, 2]
| Moved to state [0, 3] | Plan so far ['left', 'left', 'up', 'right'] |
Exploring action 'left' to move to state [0, 2] | State [0, 2] is invalid |
Exploring action 'right' to move to state [0, 4] | State [0, 4] is invalid |
Exploring action 'up' to move to state [-1, 3] | State [-1, 3] is invalid |
Exploring action 'down' to move to state [1, 3] | State [1, 3] is invalid |
Exploring action 'up' to move to state [-1, 2] | State [-1, 2] is invalid |
Exploring action 'down' to move to state [1, 2] | State [1, 2] is invalid |
Exploring action 'down' to move to state [2, 2] | State [2, 2] is valid |
Taking action 'down' from state [1, 2] | Moved to state [2, 2] | Plan so far
['left', 'left', 'down'] | Exploring action 'left' to move to state [2, 1] |
State [2, 1] is valid | Taking action 'left' from state [2, 2] | Moved to
state [2, 1] | Plan so far ['left', 'left', 'down', 'left'] | Exploring action
'left' to move to state [2, 0] | State [2, 0] is invalid | Exploring action
'right' to move to state [2, 2] | State [2, 2] is invalid | Exploring action
'up' to move to state [1, 1] | State [1, 1] is invalid | Exploring action
'down' to move to state [3, 1] | State [3, 1] is valid | Taking action 'down'
from state [2, 1] | Moved to state [3, 1] | Plan so far ['left', 'left',
'down', 'left', 'down'] | Exploring action 'left' to move to state [3, 0] |
State [3, 0] is valid | Taking action 'left' from state [3, 1] | Moved to
state [3, 0] | Plan so far ['left', 'left', 'down', 'left', 'down', 'left'] |
Exploring action 'left' to move to state [3, -1] | State [3, -1] is invalid |
Exploring action 'right' to move to state [3, 1] | State [3, 1] is invalid |
Exploring action 'up' to move to state [2, 0] | State [2, 0] is invalid |
Exploring action 'down' to move to state [4, 0] | State [4, 0] is invalid |
Exploring action 'right' to move to state [3, 2] | State [3, 2] is valid |
Taking action 'right' from state [3, 1] | Moved to state [3, 2] | Plan so far
['left', 'left', 'down', 'left', 'down', 'right'] | Goal state [3, 2] reached!
```

**Controller Prompt for Maze**

You are in a 2d maze of dimensions **5** and **5** and some of the cells have walls. The walls are placed in cells [[3, 4], [0, 4], [3, 3], [1, 1], [2, 3], [0, 0], [2, 0], [4, 0], [1, 0], [2, 4]].

Given a start and a goal state, your task is to first generate a set of subgoals that can then be solved to generate the optimal plan between the states. The optimal plan is one that has the minimum number of steps. The list of permissible actions that you can take at any given cell are ['left', 'right', 'up', 'down'].

Generate the subgoals for generating a plan from [1, 4] to [3, 2].

**Controller Output (Sub-goal Decomposition + Easy / Hard Annotations) for Maze**

The walls between [1, 4] to [3, 2] are at [2, 3], [2, 4], [3, 3], [3, 4] which are a total of **4** walls. Based on this, the plan should be broken down into 2 sub-goals:

1. Start State: [1, 4] and Goal State: [2, 2]. Planning for this subgoal is **hard** and can be solved **slow** using **system 2**
2. Start State: [2, 2] and Goal State: [3, 2]. Planning for this subgoal is **easy** and can be solved **fast** using **system 1**

---

**System-1 / System-2 Prompt for Blocksworld**

```
Your task is to generate a plan for a blocksworld problem given an initial state and a
goal state.

The initial state:
The hand is empty.
C is on the table. B is on C. E is on B. E is clear.
D is on the table. A is on D. A is clear.

The goal is:
A is on the table. C is on A. B is on C. B is clear.
D is on the table. E is on D. E is clear.
```

---

**System-1 Output for Blocksworld (#States-Explored=6)**

```
Move E from B to table | Move A from D to table | Move E from table to D | Move B from C
to table | Move C from table to A | Move B from table to C
```

---

**System-2 (A*) Output for Blocksworld (#States-Explored=28)**

```
Moved to state -> Stack 1: [C, B, E], Stack 2: [D, A] | Plan so far [] | Exploring action
Move E from B to A | New state -> Stack 1: [C, B], Stack 2: [D, A, E] | Exploring action
Move E from B to table | New state -> Stack 1: [C, B], Stack 2: [D, A], Stack 3: [E] |
Exploring action Move A from D to table | New state -> Stack 1: [A], Stack 2: [C, B, E],
Stack 3: [D] | Exploring action Move B from C to E | New state -> Invalid | Taking action
Move E from B to table | Moved to state -> Stack 1: [C, B], Stack 2: [D, A], Stack 3: [E]
| Plan so far [Move E from B to table] | Exploring action Move B from C to table | New
state -> Stack 1: [B], Stack 2: [C], Stack 3: [D, A], Stack 4: [E] | Exploring action
Move A from D to table | New state -> Stack 1: [A], Stack 2: [C, B], Stack 3: [D], Stack
4: [E] | Exploring action Move D from table to E | New state -> Invalid | Exploring
action Move A from D to E | New state -> Stack 1: [C, B], Stack 2: [D], Stack 3: [E, A] |
Exploring action Move C from table to E | New state -> Invalid | Taking action Move A
from D to table | Moved to state -> Stack 1: [A], Stack 2: [C, B], Stack 3: [D], Stack 4:
[E] | Plan so far [Move E from B to table | Move A from D to table] | Exploring action
Move B from C to E | New state -> Stack 1: [A], Stack 2: [C], Stack 3: [D], Stack 4: [E,
B] | Exploring action Move C from table to D | New state -> Invalid | Exploring action
Move E from table to D | New state -> Stack 1: [A], Stack 2: [C, B], Stack 3: [D, E] |
Exploring action Move B from C to table | New state -> Stack 1: [A], Stack 2: [B], Stack
3: [C], Stack 4: [D], Stack 5: [E] | Exploring action Move C from table to A | New state
-> Invalid | Taking action Move E from table to D | Moved to state -> Stack 1: [A], Stack
2: [C, B], Stack 3: [D, E] | Plan so far [Move E from B to table | Move A from D to table
| Move E from table to D] | Exploring action Move B from C to A | New state -> Stack 1:
[A, B], Stack 2: [C], Stack 3: [D, E] | Exploring action Move B from C to table | New
state -> Stack 1: [A], Stack 2: [B], Stack 3: [C], Stack 4: [D, E] | Exploring action
Move C from table to B | New state -> Invalid | Exploring action Move A from table to B |
New state -> Stack 1: [C, B, A], Stack 2: [D, E] | Exploring action Move C from table to
A | New state -> Invalid | Taking action Move B from C to table | Moved to state -> Stack
1: [A], Stack 2: [B], Stack 3: [C], Stack 4: [D, E] | Plan so far [Move E from B to table
| Move A from D to table | Move E from table to D | Move B from C to table] | Exploring
action Move A from table to B | New state -> Stack 1: [B, A], Stack 2: [C], Stack 3: [D,
E] | Exploring action Move B from table to C | New state -> Stack 1: [A], Stack 2: [C,
B], Stack 3: [D, E] | Exploring action Move D from table to E | New state -> Invalid |
Exploring action Move C from table to A | New state -> Stack 1: [A, C], Stack 2: [B],
Stack 3: [D, E] | Exploring action Move D from table to B | New state -> Invalid | Taking
action Move C from table to A | Moved to state -> Stack 1: [A, C], Stack 2: [B], Stack 3:
[D, E] | Plan so far [Move E from B to table | Move A from D to table | Move E from table
to D | Move B from C to table | Move C from table to A] | Exploring action Move C from A
to table | New state -> Stack 1: [A], Stack 2: [B], Stack 3: [C], Stack 4: [D, E] |
Exploring action Move A from table to C | New state -> Invalid | Exploring action Move B
from table to C | New state -> Stack 1: [A, C, B], Stack 2: [D, E] | Exploring action
Move E from D to table | New state -> Stack 1: [A, C], Stack 2: [B], Stack 3: [D], Stack
4: [E] | Taking action Move B from table to C | Moved to state -> Stack 1: [A, C, B],
Stack 2: [D, E] | Plan so far [Move E from B to table | Move A from D to table | Move E
from table to D | Move B from C to table | Move C from table to A | Move B from table to
C] | Reached goal state!
```

**Controller Prompt for Blocksworld**

```
You are given a blocksworld problem and its initial state, and goal state.
Your task is to first generate a set of subgoals that can then be solved to
generate the plan between the states.

The initial state:
The hand is empty.
C is on the table. B is on C. E is on B. E is clear.
D is on the table. A is on D. A is clear.

The goal is:
A is on the table. C is on A. B is on C. B is clear.
D is on the table. E is on D. E is clear.
```

**Controller Output (Sub-goal Decomposition + Easy / Hard Annotations) for Blocksworld**

```
The plan should be broken down into 2 sub-goals:
1. Start State -> Stack 1: [C, B, E], Stack 2: [D, A] and Goal State -> Stack
   1: [A], Stack 2: [C, B], Stack 3: [D, E].
Planning for this subgoal is hard and can be solved slow using system 2

2. Start State -> Stack 1: [A], Stack 2: [C, B], Stack 3: [D, E] and Goal
State -> Stack 1: [A, C, B], Stack 2: [D, E].
Planning for this subgoal is easy and can be solved fast using system 1
```

# E TABLES CORRESPONDING TO RESULT FIGURES

In the tables below, we report the accuracies corresponding to all the plots in the paper.

Table 3: Comparison of System-1.$x$ Planner ($x = 0.5$) with all baselines on Maze Navigation (corresponding to Fig. 3(a)). The blue rows indicate plan validity obtained by all methods when #States-Explored is matched to that of System 1.5 with sub-goals. The green rows indicate plan validity obtained when #States-Explored is maximized. All tables, henceforth, will use the same color coding.

|  | Plan Validity | #States-Explored |
|---|---|---|
| System-1 | 48.7 | 3.1 |
| System-2 (truncated) | 12.2 | 4.8 |
| System-2 (truncated) | 19.7 | 9.9 |
| System-2 (truncated) | 37.2 | 13.6 |
| System-2 (truncated) | 37.2 | 14.8 |
| System-2 (truncated) | 63.7 | 19.8 |
| System-2 (default) | 93.7 | 24.4 |
| System-1.5 w/o sub-goal (truncated) | 33.5 | 4.8 |
| System-1.5 w/o sub-goal (truncated) | 38.2 | 9.7 |
| System-1.5 w/o sub-goal (truncated) | 50.0 | 13.4 |
| System-1.5 w/o sub-goal (truncated) | 62.2 | 15.0 |
| System-1.5 w/o sub-goal (default) | 79.5 | 17.4 |
| System-1.5 w/o sub-goal (test-time controlled) | 83.7 | 20.0 |
| System-1.5 (w/o sub-goal) (test-time controlled) | 93.7 | 24.4 |
| System-1.5 w/ sub-goal (truncated) | 30.5 | 4.5 |
| System-1.5 w/ sub-goal (truncated) | 56.2 | 10.0 |
| System-1.5 w/ sub-goal (default) | **70.4** | 13.6 |
| System-1.5 w/ sub-goal (test-time controlled) | 82.2 | 19.9 |
| System-1.5 w/ sub-goal (test-time controlled) | 92.2 | 24.4 |
| System-1.5 w/ sub-goal (test-time controlled) | **96.7** | 27.3 |

Table 4: Comparison of System-1.$x$ Planner ($x = 0.5$) with all baselines on Blocksworld (corresponding to Fig. 3(b)).

|  | Plan Validity | #States-Explored |
|---|---|---|
| System 1 | 9.0 | 4.5 |
| System 2 (truncated) | 0.0 | 9.8 |
| System 2 (truncated) | 0.0 | 19.9 |
| System 2 (truncated) | 13.0 | 29.3 |
| System 2 (truncated) | 27.0 | 38.2 |
| System 2 (default) | 28.0 | 55.5 |
| System-1.5 w/o sub-goal (truncated) | 0.0 | 9.8 |
| System-1.5 w/o sub-goal (truncated) | 0.0 | 19.9 |
| System-1.5 w/o sub-goal (truncated) | 13.0 | 29.3 |
| System-1.5 w/o sub-goal (truncated) | 27.0 | 38.0 |
| System-1.5 w/o sub-goal (default) | 28.0 | 55.1 |
| System-1.5 w/ sub-goal (truncated) | 1.5 | 9.6 |
| System-1.5 w/ sub-goal (truncated) | 17.5 | 20.0 |
| System-1.5 w/ sub-goal (truncated) | 25.0 | 30.0 |
| System-1.5 w/ sub-goal (default) | 25.0 | 38.3 |
| System-1.5 w/ sub-goal (test-time controlled) | 26.0 | 53.5 |

Table 5: Comparison of Neuro-symbolic System-1.$x$ Planner ($x = 0.5$) with all baselines on Maze Navigation (corresponding to Fig. 4).

| | Plan Validity | #States-Explored |
|---|---|---|
| A$^*$ (truncated) | 12.5 | 4.8 |
| A$^*$ (truncated) | 22.5 | 9.8 |
| A$^*$ (truncated) | 31.0 | 11.3 |
| A$^*$ (truncated) | 42.0 | 14.5 |
| A$^*$ (truncated) | 67.7 | 19.8 |
| A$^*$ (default) | **100.0** | 22.4 |
| System-1.5 w/o sub-goal (truncated) | 33.0 | 4.8 |
| System-1.5 w/o sub-goal (truncated) | 40.2 | 9.8 |
| System-1.5 w/o sub-goal (truncated) | 45.7 | 11.5 |
| System-1.5 w/o sub-goal (truncated) | 65.7 | 14.9 |
| System-1.5 w/o sub-goal (default) | 84.2 | 16.5 |
| System-1.5 w/o sub-goal (test-time controlled) | 94.0 | 20.0 |
| System-1.5 w/o sub-goal (test-time controlled) | **100.0** | 22.4 |
| System-1.5 w/ sub-goal (truncated) | 30.5 | 4.5 |
| System-1.5 w/ sub-goal (truncated) | 58.7 | 10.0 |
| System-1.5 (default) | **70.5** | 11.6 |
| System-1.5 w/ sub-goal (test-time controlled) | 76.0 | 14.8 |
| System-1.5 w/ sub-goal (test-time controlled) | 89.7 | 19.9 |
| System-1.5 w/ sub-goal (test-time controlled) | 99.2 | 23.9 |

Table 6: Comparison of System-1.$x$ Planner ($x = 0.75$) with all baselines on Maze Navigation (corresponding to Fig. 5).

| | Plan Validity | #States-Explored |
|---|---|---|
| System-2 (truncated) | 12.2 | 4.8 |
| System-2 (truncated) | 19.7 | 9.9 |
| System-2 (truncated) | 37.2 | 14.8 |
| System-2 (truncated) | 47.0 | 16.6 |
| System-2 (truncated) | 63.7 | 19.8 |
| System-2 (default) | 93.7 | 24.4 |
| System-1.75 w/o sub-goal (truncated) | 24.2 | 4.6 |
| System-1.75 w/o sub-goal (truncated) | 29.2 | 9.8 |
| System-1.75 w/o sub-goal (truncated) | 44.5 | 14.6 |
| System-1.75 w/o sub-goal (default) | 59.7 | 16.6 |
| System-1.75 w/o sub-goal (default) | 86.5 | 20.3 |
| System-1.75 w/o sub-goal (test-time controlled) | 93.7 | 24.4 |
| System-1.75 w/ sub-goal (truncated) | 23.0 | 4.9 |
| System-1.75 w/ sub-goal (truncated) | 38.7 | 9.6 |
| System-1.75 w/ sub-goal (truncated) | 66.7 | 15.0 |
| System-1.75 w/ sub-goal (default) | **75.7** | 16.6 |
| System-1.75 w/ sub-goal (test-time controlled) | 82.0 | 19.9 |
| System-1.75 w/ sub-goal (test-time controlled) | 91.2 | 24.4 |
| System-1.75 w/ sub-goal (test-time controlled) | **96.7** | 26.8 |

Table 7: Comparison of System-1.$x$ Planner ($x = 0.5$ and BFS) with all baselines on Maze Navigation (corresponding to Fig. 6(a)).

|  | Plan Validity | #States-Explored |
|---|---|---|
| System-2 (truncated) | 11.7 | 9.4 |
| System-2 (truncated) | 28.0 | 17.4 |
| System-2 (truncated) | 28.0 | 19.9 |
| System-2 (truncated) | 56.2 | 29.7 |
| System-2 (default) | 91.2 | 39.8 |
| System-1.5 w/o sub-goal (truncated) | 35.2 | 9.6 |
| System-1.5 w/o sub-goal (truncated) | 45.0 | 17.2 |
| System-1.5 w/o sub-goal (truncated) | 52.5 | 19.9 |
| System-1.5 w/o sub-goal (default) | 77.2 | 26.9 |
| System-1.5 w/o sub-goal (test-time controlled) | 81.2 | 30.0 |
| System-1.5 w/o sub-goal (test-time controlled) | 91.2 | 39.8 |
| System-1.5 w/ sub-goal (truncated) | 43.0 | 9.8 |
| System-1.5 w/ sub-goal (default) | **67.0** | 17.4 |
| System-1.5 w/ sub-goal (test-time controlled) | 78.5 | 29.7 |
| System-1.5 w/ sub-goal (test-time controlled) | 86.5 | 39.9 |
| System-1.5 w/ sub-goal (test-time controlled) | **91.7** | 45.4 |

Table 8: Comparison of System-1.$x$ Planner ($x = 0.5$ and DFS) with all baselines on Maze Navigation (corresponding to Fig. 6(b)).

|  | Plan Validity | Avg States |
|---|---|---|
| System-2 (truncated) | 11.7 | 4.7 |
| System-2 (truncated) | 23.2 | 9.7 |
| System-2 (truncated) | 42.0 | 14.5 |
| System-2 (truncated) | 69.7 | 19.8 |
| System-2 (truncated) | 81.2 | 25.0 |
| System-2 (default) | **83.5** | 30.4 |
| System-1.5 w/o sub-goal (truncated) | 36.0 | 4.7 |
| System-1.5 w/o sub-goal (truncated) | 50.7 | 9.8 |
| System-1.5 w/o sub-goal (truncated) | 73.2 | 15.0 |
| System-1.5 w/o sub-goal (default) | 74.0 | 18.7 |
| System-1.5 w/o sub-goal (test-time controlled) | 76.3 | 19.8 |
| System-1.5 w/o sub-goal (test-time controlled) | 81.2 | 25.0 |
| System-1.5 w/o sub-goal (test-time controlled) | **83.5** | 30.4 |
| System-1.5 (truncated) | 40.2 | 4.5 |
| System-1.5 (default) | **62.3** | 10.0 |
| System-1.5 (test-time controlled) | 76.3 | 14.9 |
| System-1.5 (test-time controlled) | 79.7 | 20.0 |
| System-1.5 (test-time controlled) | 82.7 | 33.6 |

Table 9: Comparison of System-1.$x$ Planner with and without sliding window decomposition (corresponding to Fig. 7).

|  | Plan Validity | #States-Explored |
| --- | --- | --- |
| System-1.5 w/o Sliding Window (truncated) | 27.5 | 4.6 |
| System-1.5 w/o Sliding Window (truncated) | 45.2 | 10.0 |
| System-1.5 w/o Sliding Window (default) | 67.2 | 17.0 |
| System-1.5 w/o Sliding Window (test-time controlled) | 72.1 | 20.0 |
| System-1.5 w/o Sliding Window (test-time controlled) | 77.2 | 25.0 |
| System-1.5 w/o Sliding Window (test-time controlled) | 83.2 | 30.0 |
| System-1.5 w/o Sliding Window (test-time controlled) | 90.0 | 35.3 |
| System-1.5 w/ Sliding Window (truncated) | 30.5 | 4.5 |
| System-1.5 w/ Sliding Window (truncated) | 56.2 | 10.0 |
| System-1.5 w/ Sliding Window (default) | **70.4** | 13.6 |
| System-1.5 w/ Sliding Window (test-time controlled) | 82.2 | 19.9 |
| System-1.5 w/ Sliding Window (test-time controlled) | 92.2 | 24.4 |
| System-1.5 w/ Sliding Window (test-time controlled) | **96.7** | 27.3 |

Table 10: Comparison of two hardness functions (Manhattan Distance and #Obstacles) for Maze Navigation (corresponding to Fig. 8).

|  | Plan Validity | #States-Explored |
| --- | --- | --- |
| System-1.5 w/o sub-goal (#Obstacles) | 33.5 | 4.8 |
| System-1.5 w/o sub-goal (#Obstacles) | 38.2 | 9.7 |
| System-1.5 w/o sub-goal (#Obstacles) | 50.0 | 13.4 |
| System-1.5 w/o sub-goal (#Obstacles) | 62.2 | 15.0 |
| System-1.5 w/o sub-goal (#Obstacles) | 79.5 | 17.4 |
| System-1.5 w/o sub-goal (#Obstacles) | 83.7 | 20.0 |
| System-1.5 w/o sub-goal (#Obstacles) | 93.7 | 24.4 |
| System-1.5 w/o sub-goal (Manhattan) | 31.7 | 4.8 |
| System-1.5 w/o sub-goal (Manhattan) | 54.5 | 15.0 |
| System-1.5 w/o sub-goal (Manhattan) | 35.2 | 10.0 |
| System-1.5 w/o sub-goal (Manhattan) | 80.0 | 18.8 |
| System-1.5 w/o sub-goal (Manhattan) | 81.5 | 19.9 |
| System-1.5 w/o sub-goal (Manhattan) | 93.7 | 24.4 |

