# OpenReview forum: "System 1.x: Learning to Balance Fast and Slow Planning with Language Models"
_ICLR.cc/2025/Conference — ICLR 2025 Poster_

### Official Review · Reviewer_s6YJ · 2024-10-19

**Soundness:** 1
**Presentation:** 2
**Contribution:** 1
**Rating:** 1
**Confidence:** 5

**Summary:**

This paper attempts to create a mixed-mode planning system that both immediately generates plans without deliberate state-space search, and does deliberate state space search.

However, this paper seems deliberately misleading in terms of what it claims, namely the improvements over A*.

**Strengths:**

In general, the goal of creating a neurosymbolic planning system is a very good one. However, the one presented in this paper is preliminary.

I really like the idea of having a hybridization factor. It could be interesting to have a way to control this at inference time rather than at training time, i.e. like iterative-deeping based searches use more search budget when given more time.

**Weaknesses:**

While this work attempts to build a neurosymbolic planning system, which is a good goal, it seems like the work may not be ready for publication at ICLR.

The main results of the paper rest on a claim about the "States explored" metric, however, there are several issues with this evaluation. Primarily, computational costs of querying LLMs are not accounted for. A single pass of an LLM clearly can do a significant amount of computation. Furthermore, intermediate LLM-generated states are never checked, e.g. they could contain hallucinations, so it doesn't seem like they'd make a reasonable metric. More importantly there is simply no guaranteed correspondence between intermediate states inside of A* and reported intermediates produced by an LLM to the degree that the two can be compared meaningfully. Because of this, it is unreasonable to restrict A* to run in correspondence with the LLM output, when a full A* computation may several orders of magnitude less than a computational single pass of an LLM. Since the paper rests on the states explored metric, I highly encourage the authors to propose a better method for comparing computational costs between LLMs and A*.

Another ommission is the fact the the overall system is trained by calling A* thousands of times, which makes an efficiency argument extremely questionable. Regardless of the current state of such an argument, I do buy into the general idea that mixed-system planning algorithm could be more efficient than a system-2 one and more accurate than a system-1 one, but I think the authors would need to fairly evaluate this to make progress on building such an algorithm.

The other main metric used in the paper is "valid plan accuracy". Such a metric is not meaningful when using systems with guarantees, namely A*, which is guaranteed to find optimal plans, while the proposed system has no such guarantees. Because of the false equation between A* intermediate states and LLM-output, the budget restriction is not meaningful and skews the accuracy metric significantly. For instance, if wall time budget were used instead, A* would simply have an accuracy of 100%. Reporting otherwise is potentially misleading.

The paper claims to test out-of-distribution blocksworld, but it is unclear from figure 1 which results are out of distribution testing and which are in-distribution testing. Also, the left hand side of figure 1 has a correct maximum accuracy of 100%, but the right hand side has a maximum accuracy of only 30%, which seems extremely misleading. Can you clearly label which results in Figure 1 are from in-distribution versus out-of-distribution testing and present the results in Figure 1 with a uniform scale 0-100?

"model-based" on line 144 seems misused, namely confusing parameterized models like neural networks with transition models used in planning and RL. In general, some terminology is unconventional or misused.

line 253: "we assume that they can be decomposed into sub-goals". This is a very dangerous assumption that rules out true planning problems, as often sub-goals interact. This is the case for blocksworld (but not the maze task), so since the blocksworld dataset has been filtered in this way, it no longer represents the original task and is in fact far simpler to solve [1]. The method for decomposing plans is not obvious, and the sliding window approach, while interesting, is certainly not guaranteed to decompose an arbitrary problem successfully. In the worst case, the system must always be able to fall back on System 2.

The system is given tiny datasets and (for mazes) tested within distribution. 5x5 mazes are far too small and A* can scale to much bigger problems, while it's unclear if the proposed LLM-based approach does. The relatively high system 1 performance is a good indicator that parts of the problem are being memorized -- and in the case of mazes it is clear how this is possible since the task has decomposable subgoals that can be independently memorized. It's good that out-of-distribution was considered for blocksworld, but the results section doesn't always distinguish between numbers reported from in-distribution testing and out-of-distribution testing.

In general, planning is misunderstood. By default A* is sound, complete, and produces optimal plans. In contrast, an LLM generator is not sound, not complete, and not optimal. When A* is limited by search budget, it is not the same algorithm, but still will never produce an incorrect plan, and if given enough time will produce the optimal plan. In contrast, an LLM (or system that uses one) is capable of producing incorrect plans, and has no optimality guarantee even if given infinite computation. The proposed method does not have any guarantees, wheras systems like LLM-Modulo do have soundness guarantees (but not completeness!). Please consider adding a discussion section that explicitly compares the theoretical guarantees and limitations of A* versus the proposed LLM-based approach. This could help readers better understand the tradeoffs involved and the contexts in which each approach might be most appropriate.

[1] Sussman, G. J. (1973). A computational model of skill acquisition.

See also: https://en.wikipedia.org/wiki/Sussman_anomaly

**Questions:**

In the results section for blocksworld, which numbers are from in-distribution testing and which are from out-of-distribution testing?

What guarantees does System 1.75 have?

Did you consider wall-clock time budgets? This would be a more fair comparison. Surely the authors know that querying LLMs repeatedly takes far longer than querying A*, so I'm curious why these times weren't reported in the paper?

**Details Of Ethics Concerns:**

Figure 3 seems to be cropped in a way that makes the method appear to be better than it is, and in general the reported metrics are simply not meaningful (see weaknesses).

---

> ### Author Response · Authors · 2024-11-25
> **Response to Reviewer s6YJ (Part 1)**
>
> Thank you for your review!
>
> > Figure 3 seems to be cropped in a way that makes the method appear to be better than it is
>
> Let us begin by first addressing your ethical concern. We unequivocally reject this claim as it is not a criticism of our paper but in fact calls into question the integrity of our research, which is completely unfounded and presumptuous with no evidence.
>
> First, the figure on the right (3b) is for Blocksworld and the one on the left (3a) is for Mazes. Their experimental setups are different and completely independent. For BW, as noted in Lines 300-305, we follow a OOD setup – we train on plan lengths of 1-6 and test on longer plan lengths of 7-10. The setup for Mazes is in-distribution – we both train and test on plan lengths of 1-8.
>
> Next, given this OOD setup for BW, our System-2 Planner (that is trained on search traces) obtains an accuracy of 28% (see the highest point on the green Sys-2 line to the far right; it overlaps with the violet line and hence might not be clearly visible; so you could refer to Table 4 in Appendix for exact numbers). This number is higher than the System-1 accuracy of 9% (the horizontal blue line) but expectedly, far worse than the high System-2 accuracy for Mazes (which followed an ID setup and hence, easier).
>
> So, to summarize, the maximum achievable accuracy for BW is still 100% (not 30%). We happened to choose a scale of 0-30 because our System-2 Planner achieves an accuracy of 28% and by design, System-2 represents the skyline of a System 1.x Planner (when x=1 and System 1.x becomes the same as a System-2). If it helps, we are happy to change the scale of the BW plot also to 0-100 in a future version.
>
> > While this work attempts to build a neurosymbolic planning system, which is a good goal, it seems like the work may not be ready for publication at ICLR.
>
> As pointed to in the general response, this is in fact **not** the goal of our work. As our title suggests and as is clearly stated in the abstract lines 19-20, our paper is about **LLMs** and how we can make LLMs **alone** more efficient in solving planning problems by internalizing (System-1) and externalizing (System-2) search for sub-goals based on a user-defined compute budget. The abstract and many other parts of the paper (Section 2 heading, etc) repeatedly uses the phrase **"System-1.x Planner, a framework for controllable planning with language models"** where the **"with language models"** part is important and should not be overlooked. If that is our goal, which we think we have clearly communicated in our paper, then the baselines of our hybrid System 1.x should be an **LLM-based** System-1 Planner and an **LLM-based** System-2 Planner. And if these are our baselines, which again we have compared to in our paper, States-explored is the right metric to compare against because all are LLM-based methods and States-explored directly correlates with token count and token count is what dictates the compute budget of an LLM. System-1.x outperforms both these baselines and hence, should be seen as our contribution towards making **LLMs** search more efficiently by internalizing easier sub-goals when operating under a budget.
>
> Next, let us clarify the specific point about neuro-symbolic planning and comparison to A*. Our paper, in parts, does talk about a neuro-symbolic System 1.x variant. However, again the keyword overlooked in the review is **variant**. It’s a variant because it naturally emerges from the modular design of our framework wherein we disentangled the capabilities of a slow planner, a fast planner, and a controller (so the slow planner could also be A*). In our paper, we intentionally used phrases like **"System-1.x can be converted into a neuro-symbolic variant"** (line 115) and more generally, presented the neuro-symbolic results in a separate sub-section 4.2 to emphasize the point that we are **not** trying to build the best neuro-symbolic architecture for Mazes and BW. We absolutely understand that A* alone can solve a lot of these problems (with guarantees) and that our System 1.x may not be directly comparable to A* in terms of the States-explored metric because generating the tokens for a state by an LLM requires more computation than doing so with A*. However, wall-clock time would also not be appropriate because we are trying to make LLMs better and more efficient planners and an LLM’s typical unit of compute is token.
>
> To summarize, we are happy to rephrase our neuro-symbolic part in a future version to further clarify our research goal. Regardless, System 1.x, even with its current presentation, is a novel framework for hybrid and controllable planning **with LLMs** that advances the state-of-the-art on full System-2 architectures (e.g., Searchformer, Stream-of-Search) that always make LLMs externalize search.
>
> And if we are able to successfully communicate this research goal of ours, it should help address your other questions about guarantees of a System 1.x and such.

---

> > ### Author Response · Authors · 2024-11-25
> > **Response to Reviewer s6YJ (Part 2)**
> >
> > > Another omission is the fact the the overall system is trained by calling A* thousands of times, which makes an efficiency argument extremely questionable.
> >
> > When building an LLM-based planner, we need some expert demonstrations/search trajectories to train on. The A* trajectories, in our paper, are representative of that. In general, these could come from humans or be curated in any other automated way. This is not an omission, as we write this clearly in our Problem Setup (Lines 179-187).
> >
> > > The paper claims to test out-of-distribution blocksworld, but it is unclear from figure 1 which results are out of distribution testing and which are in-distribution testing.
> >
> > Please refer to our first response about the ethical concern..
> >
> > > The method for decomposing plans is not obvious, and the sliding window approach, while interesting, is certainly not guaranteed to decompose an arbitrary problem successfully. In the worst case, the system must always be able to fall back on System 2.
> >
> > We show empirically that it works for the tasks we considered. It’s conceivable that for a different domain, one might need to update our sliding window approach or maybe even collect some human annotations for sub-goal decompositions to train the controller. However, the conceptualization of System-1.x will still remain as one of our contributions, which is that one needs a controller to balance the slow and fast planners.
> >
> > > In general, planning is misunderstood. By default A* is sound, complete, and produces optimal plans. In contrast, an LLM generator is not sound, not complete, and not optimal.
> >
> > By this logic, all past work performing planning with an LLM has also misunderstood planning. Moreover, as you point out in saying that neuro-symbolic planning is valuable, there must be some room for neural models (e.g., generators) in planning – otherwise all non-A* planning would be pointless. We would appreciate further clarification of how neuro-symbolic planning can be "a very good" goal but incorporating LLMs into planning misunderstands planning. We know that an LLM that generates search traces may hallucinate and may not have guarantees like A*. However, it should not stop us from exploring to what extent and with how much efficiency, can we use LLMs to search and solve planning problems. System 1.x takes a positive step in that direction.

---

> > ### Comment · Reviewer_s6YJ · 2024-11-25
> > **Response to Authors**
> >
> > > So, to summarize, the maximum achievable accuracy for BW is still 100% (not 30%). We happened to choose a scale of 0-30 because our System-2 Planner achieves an accuracy of 28%
> >
> > Precisely, this is misleading. The scale should be 0-100.
> >
> > Overall, I am still not convinced that the evaluation of the paper is done in good faith, or that ignoring/misrepresenting existing algorithms like A* makes sense.

---

### Official Review · Reviewer_z8RW · 2024-11-03

**Soundness:** 2
**Presentation:** 2
**Contribution:** 1
**Rating:** 3
**Confidence:** 5

**Summary:**

The paper proposes System-1.x, a hybrid planning framework using language models to balance fast, intuitive System-1 planning with deliberate, accurate System-2 planning.

**Strengths:**

System-1.x shows robust adaptability and performance across different planning tasks (e.g., Maze Navigation, Blocksworld), outperforming both purely System-1 and System-2 planners due to its effective sub-goal decomposition.

**Weaknesses:**

The related work section of the paper does not cover papers similar to this work. To cite a few papers that should have been used as baselines to compare System 1.x against -
[1] Katz, M., Kokel, H., Srinivas, K., & Sohrabi, S. (2024). Thought of Search: Planning with Language Models Through The Lens of Efficiency. In The First Workshop on System-2 Reasoning at Scale, NeurIPS'24.
[2] Fabiano, F., Pallagani, V., Ganapini, M. B., Horesh, L., Loreggia, A., Murugesan, K., ... & Srivastava, B. (2023, December). Plan-SOFAI: A Neuro-Symbolic Planning Architecture. In Neuro-Symbolic Learning and Reasoning in the era of Large Language Models.
[3] Webb, T., Mondal, S. S., Wang, C., Krabach, B., & Momennejad, I. (2023). A Prefrontal Cortex-inspired Architecture for Planning in Large Language Models. arXiv preprint arXiv:2310.00194.
Refer to the questions for more weaknesses.

**Questions:**

1. In the experimental setup, the authors mention that the hybridization factor x allows System-1.x to control the degree of System-1 versus System-2 planning. How sensitive is this hybridization factor to changes in task complexity? Were there significant trade-offs observed with larger values of x in different domains?
2. In discussing Blocksworld’s out-of-distribution generalization, the paper mentions that System-1.x effectively utilizes System-1 for ID sub-goals while reserving System-2 for OOD sub-goals. Could the authors explain how the controller identifies which sub-goals are in-distribution vs. out-of-distribution during inference?
3. The authors claim that System-1.x can match or exceed a purely System-2 Planner’s accuracy with fewer states explored. What specific methods were used to prevent overfitting to the training traces, especially for tasks requiring extensive state exploration?
4. Why is there no baseline comparison against other fast and slow architectures used for solving planning problems? It is not clear to me what is the advantage of using this approach against architectures like Plan-SOFAI that are both optimal and resource-efficient compared to System 2 solvers for planning tasks.

---

> ### Author Response · Authors · 2024-11-25
> **Response to Reviewer z8RW**
>
> Thank you for your review!
>
> > How sensitive is this hybridization factor to changes in task complexity? Were there significant trade-offs observed with larger values of x in different domains?
>
> With larger values of x, System 1.x will start approaching a full System-2. How fast (i.e., the slope of the curve) will depend on the complexity of the end task and the exact problem setup. For example, in our experiments with Mazes, we find that our System-1.5 achieves an accuracy of 70%, still a fair way short of the maximum 94% obtained by a full System-2 Planner (see Lines 370-381). Such high performance from a System-2 Planner is expected because we train and test on a similar distribution of mazes and plan lengths. On the other hand, for Blocksworld, we test on longer plan lengths than what we train on and thus, even a full System-2 Planner is able to solve a considerably smaller number of problems. In such a scenario, System 1.5 obtains an accuracy of 25%, quite close to the maximum 28% obtained by a System-2 Planner.
>
> In summary, for certain domains, where it is possible to internalize a lot of the sub-goals, a System-1.x with even smaller values of x can get us quite close to the maximum System-2 performance whereas in other harder domains, the model might need to externalize its planning a lot more, thus requiring a larger value of x.
>
> > Could the authors explain how the controller identifies which sub-goals are in-distribution vs. out-of-distribution during inference?
>
> Note that our controller is a fine-tuned LLM trained with certain heuristics/hardness functions as features. The heuristic used in Blocksworld is a distance measure based on the number of mismatches between the start and the goal state. Simply put, a trained model with such a heuristic would learn to map a greater number of mismatches to System-2 while using System-1 when the mismatches are less. Now in Blocksworld, we evaluate our model on much longer plan lengths than what we train on and plan length generally would be expected to correlate with the number of mismatches (our hardness function). Hence, a trained controller with such a hardness function would assign System-2 to goals/sub-goals with longer plan lengths, effectively translating to OOD.
>
> > What specific methods were used to prevent overfitting to the training traces, especially for tasks requiring extensive state exploration?
>
> We diversify our training data generation process by having samples with different plan lengths, varying number of obstacles (for mazes) and varying number of blocks (for blocksworld) so that the model does not overfit to one particular configuration. Additionally, for blocksworld, we evaluate on much longer plans (up to 10) to test generalization.
>
> > Why is there no baseline comparison against other fast and slow architectures used for solving planning problems?
>
> Thank you for the pointers, these are interesting studies that we’ll be happy to refer to in a future version. However, we don’t think these studies should be our baselines because our goals are separate. In particular, our goal is to study how we can leverage LLMs **alone** to solve planning problems while being able to reduce their computational cost when trained on long search traces (like Stream-of-Search/Searchformer do and make up our baselines). With that goal in mind, we point to the fact that all our components in System 1.x are fine-tuned LLMs with the same base architecture. If there are prior slow&fast architectures that also rely **only on LLMs** to solve planning problems, we would appreciate any citations and would be happy to compare to them.
>
> This goal of ours is completely different from the following – what’s the best (neuro-symbolic) architecture for solving Mazes and Blockworld. We understand that A* alone can solve a lot of these Maze and Blocksworld problems but that should not stop us from studying whether LLMs **alone** can do so too and in what capacity/using how much compute and whether that compute can be intelligently balanced. This makes LLM-based System-1 and System-2 planners our baselines because they respectively make up the baseline and the skyline of a System 1.x Planner.
>
> That said, our paper (in parts) also talks about a neuro-symbolic System 1.x which is when we compare it to A*. We do so for the sake of completeness and also because our modular design (with disentangled System-1/2/controller) allows us to have both LLM-based as well as symbolic System-2 solvers like A*. However, the goal of System 1.x is to build a **purely LLM-based** planning system that learns to balance deliberate/externalized and non-deliberate/internalized planning by intelligently allocating a user-defined compute budget. We believe that our method, baselines, and experimental design adequately address that research goal.

---

### Official Review · Reviewer_vcqi · 2024-11-04

**Soundness:** 3
**Presentation:** 3
**Contribution:** 2
**Rating:** 6
**Confidence:** 4

**Summary:**

This paper introduces System-1.x which acts as a hybrid System 1 / System 2 planner that outperforms fully System 1 and 2 systems, an A* planner, and a System-1.x variant without subgoal decomposition. This paper contributes a method for collecting data to train a System-1.x system through the introduction of domain-specific hardness functions to determine which of the collected samples should be annotated for System-1 (easy samples) or System-2 (hard samples).

**Strengths:**

$Originality:$ This paper introduces a new method to collect data to finetune a hybrid System 1/2 system whose hybridization factor x can be customized during train and test time.

$Quality:$ The figures and methodology are well-done and the experiments cover relevant baselines and ablations.

$Clarity:$ The paper is well-written and clear.

$Significance:$ Constructing LLM planners that can outperform classical planning approaches with minimal exploration is very important in domains where querying the world model can be expensive (e.g. real world systems like robotics).

**Weaknesses:**

The introduction makes the following claim

"System-1.x is a fully LLM-based neural planner ... not relying on any external solvers or verifiers... it has an advantage over inference-time methods that rely on symbolic planners which may not exist for certain domains."

However, System-1.x strongly rely on the presence of a hardness function, which is essentially a heuristic function. This assumption is too strong and cannot simply be listed as a limitation of this approach; the domains that are tested on could be purely solved using classical planners like A* with access to the hardness (heuristic) function and a valid plan would be retrieved (regardless of consistency or admissibility of the hardness function) much faster than any LLM approach in terms of inference time.

To improve this paper, evaluating on environments where classical planners cannot be trivially applied would give more credibility to this LLM approach. A user-specified hardness function cannot be provided for the environment since a classical planner would be able to utilize this along with the other environment information provided to the System-1.x planner and quickly find a valid plan.

One potential idea is utilizing an LLM to provide a hardness value for a pair of states as other works have done, using an LLM to approximate a heuristic function [1,2,3]. This would require some new experiments to show the effectiveness of the hardness function and would necessitate a new approach other than the sliding window mechanism to be token-efficient.

This is a very important problem to solve in this space and I wish the authors the best of luck.

[1] Large Language Models as Commonsense Knowledge for Large-Scale Task Planning (Zhao et. al 2023)

[2] Reasoning with Language Model is Planning with World Model (Hao et. al 2023)

[3] Toolchain*: Efficient Action Space Navigation in Large Language Models with A* Search (Zhuang et. al 2023)

**Questions:**

- You make the case that System-1.x is advantageous over inference-time methods relying on symbolic planners since they may not exist for certain domains, yet System-1.x relies on defining a hardness function, which is essentially a heuristic function. How could System-1.x be leveraged in new domains where a user is not a domain expert?
- In addition, it is non-trivial to construct hardness functions for multi-task domains like Alfworld [1], Robotouille [2] or Mini-BEHAVIOR [3]. How would System-1.x be applied to such domains?
- There appears to be a tradeoff in Figure 3 with accuracy and states explored with Sys-1.5 with and without subgoals. Sys-1.5 with subgoals achieves higher performance with less states explored but as the state exploration budget increases, no subgoals is better. Why is that?

[1] Alfworld: Aligning text and embodied environments for interactive learning (Shridhar et. al 2020)

[2] Demo2Code: From Summarizing Demonstrations to Synthesizing Code via Extended Chain-of-Thought (Wang et. al 2023)

[3] Mini-BEHAVIOR: A Procedurally Generated Benchmark for Long-horizon Decision-Making in Embodied AI (Jin et. al 2023)

---

> ### Author Response · Authors · 2024-11-25
> **Response to Reviewer vcqi (Part 1)**
>
> Thank you for your review and for acknowledging the originality and significance of our work.
>
> > However, System-1.x strongly rely on the presence of a hardness function, which is essentially a heuristic function. This assumption is too strong and cannot simply be listed as a limitation of this approach;
>
> It is important to note that the reliance on a hardness function is only for **automatically generating training data for the controller**. Our controller is still a fine-tuned LLM that may internally use some of its pre-trained knowledge to make these hardness/sub-goal predictions (as an example, see the Controller output in Page 22). In domains where such training data is readily available, the LLM can be directly fine-tuned on that data without relying on any hardness function.
>
> In this regard, your suggestion on using an LLM to explicitly generate hardness estimates is interesting. However, the focus of our paper was to propose a general 1.x framework that works flexibly with different choices of System 1/2 and the controller. It is definitely possible to improve upon these components (e.g., by making the LLM also estimate hardness) and we hope that future work can build on top of our framework.
>
> > To improve this paper, evaluating on environments where classical planners cannot be trivially applied would give more credibility to this LLM approach.
>
> This is a well-received point. However, the primary baseline of System 1.x is a System-2 LLM planner that **always** does deliberate search (in token-space) and hence hard to implement on tasks outside of the scope of standard planning. Two prior works in this space are our baselines  (Searchformer [1] and Stream-of-Search [2], cited in our paper) , which also experimented with tasks such as Maze Navigation and games like Countdown that one could argue can also be solved using A*. Hence it is important to clarify the goal of our study – it is to understand whether LLMs can be used for search and where such (expensive) search in the token-space can be bypassed without affecting accuracy (thus, System 1.x). This goal is different from whether these problems can be solved without using an LLM (e.g., by using A*), the answer to which is of course yes. While making LLMs search in more complex environments is definitely the long-term goal, we must note that it is challenging because of the large action space leading to really long trajectories and the consequent computational challenges (something that, in our opinion, needs an entirely separate study). Thus, techniques such as our System 1.x which help reduce computational overhead by using System-1 wherever possible, should be seen as a step forward in that direction.
>
> [1] Beyond A*: Better Planning with Transformers via Search Dynamics Bootstrapping, Lenhert et al.
>
> [2] Stream of Search (SoS): Learning to Search in Language, Gandhi et al.
>
> >  How could System-1.x be leveraged in new domains where a user is not a domain expert?
>
> System 1.x is a framework for hybrid planning with LLMs. This means that applying it to a new domain requires training a System 1 Planner, a System 2 Planner, and a Controller. The System 1 and 2 Planners can be trained with only access to some exploration/search trajectories, which does not necessarily require a domain expert. The controller will be used for sub-goal decompositions and unless one wants to directly prompt an LLM to do so (which may work well, we don’t know), it should be trained with some sub-goal annotations. These annotations could either come from humans or be generated synthetically/automatically by using some domain-knowledge in the form of heuristics as hardness functions (like we do). It’s important to stress that these heuristics need not be perfect because as part of fine-tuning, we still want to leverage an LLM’s pre-trained knowledge while also teaching it to use some useful features for measuring hardness (e.g., learning to count obstacles or manhattan distance in Mazes – see our examples in the Appendix).

---

> > ### Author Response · Authors · 2024-11-25
> > **Response to Reviewer vcqi (Part 2)**
> >
> > > There appears to be a tradeoff in Figure 3 with accuracy and states explored with Sys-1.5 with and without subgoals. Sys-1.5 with subgoals achieves higher performance with less states explored but as the state exploration budget increases, no subgoals is better.
> >
> > We would first like to clarify the behavior of a "with sub-goal" and "without sub-goal" System 1.x. The "with sub-goal" version will either solve a problem entirely using System-1 or with sub-goal decompositions of System-1+System-2. The "without sub-goal" version will either solve a problem entirely using System-1 or entirely using System-2. This essentially means that when the budget is lower, a System that solves an entire problem with System-2 (i.e., without sub-goals) will not be able to complete its search, be truncated, and thus, not be able to return a plan. On the other hand, decomposition into sub-goals makes it more likely to return a valid plan, making the "with sub-goal" version better at lower budgets.
> >
> > Now, as the budget increases, one can afford more compute and if so, solving an entire problem with only System-2 is more accurate than solving some sub-goals using System-1 (and hence, may introduce errors). Thus, with more capacity, System-2 without sub-goal starts to become better until a point comes when all problems get solved with System-2 only (when x=1) and this basically equates to just having a System-2 planner with no hybridization.

---

> > > ### Comment · Reviewer_vcqi · 2024-11-26
> > >
> > > I thank the authors for their responses and the general response which have clarified a lot and I have increased my score to a 6.
> > >
> > > My understanding is that hardness functions would need to be implemented for new domains which may be cumbersome in the case of multi-task environments as mentioned before. While not the focus of this work, this is a real painpoint that the community will encounter when attempting to engage with this work. I would suggest using standard heuristics (https://www.fast-downward.org/Doc/Evaluator) to streamline this process and make engaging easier.

---

### Official Review · Reviewer_eYzS · 2024-11-04

**Soundness:** 2
**Presentation:** 3
**Contribution:** 2
**Rating:** 5
**Confidence:** 4

**Summary:**

This work proposes a hybrid planning system that combines a large language model (LLM) fine-tuned to act as either a System-1 or System-2 planner, with a Controller that decomposes tasks into sub-goals based on difficulty. The Controller generates a meta-plan by assigning "easy" sub-goals to System-1 for faster solutions and "hard" sub-goals to System-2 for more deliberate, search-based solutions. The final plan is constructed by concatenating sub-plans from both systems. However, it faces limitations due to compounding errors across systems, particularly as task complexity grows.

**Strengths:**

An innovative feature of this hybrid approach, which combines quick solutions from System-1 with thorough search-based solutions from System-2, is a user-controlled parameter x (between 0 and 1) that adjusts how much System-2 is used compared to System-1. This allows users to balance speed and accuracy based on the task’s needs.

**Weaknesses:**

1. A significant limitation of this approach is its dependency on multiple components performing flawlessly: System-1 must solve the easy sub-goals without hallucinating, System-2 must solve the harder sub-goals using search trajectories accurately, and the Controller must correctly decompose and assign tasks between the two systems. This compounded reliance on error-free performance from each component raises concerns about the approach’s reliability, especially as problem complexity increases. For even moderately difficult tasks, or as the number of objects scales, this method may struggle to maintain effectiveness, as errors are likely to accumulate across the system.
2. A limitation of this hybrid approach is that selecting the x value too greedily can hurt performance and increase training computation costs. Meanwhile, setting x too high leads to more thorough searches by System-2, which increases compute requirements and makes the approach similar to existing methods like fine-tuning LLMs for planning or SOFAI architectures [1].
3. For System-2, the hardness function should be explicitly defined for each specific domain in which the planner operates. Without domain-specific definitions, the hardness function risks being too generalized, leading to inaccurate difficulty assessments that don’t align with the unique challenges of each task.

**Questions:**

1. The term plan validity should not be used interchangeably with plan accuracy. Validity is a binary concept that indicates whether a plan meets all required conditions or constraints (i.e., valid or invalid), whereas accuracy implies a measure of correctness or closeness to an ideal solution, which can vary in degree.
2. The definition of a sub-goal as a "pair of states" adds unnecessary complexity and deviates from the usual understanding. Sub-goals are typically single intermediate states that help guide progress toward the final goal.
3. The hardness function defined for Blocksworld, in Section A.3, seems wrong. It states that if a block is not on the table and not in its goal position, an additional 1 is added to the hardness cost. However, this approach doesn't account for scenarios where a block is in the air after a pick-up action—this intermediate position may not be the goal but could be one step away from it. Adding to the hardness cost in such cases could mistakenly label states as more challenging, even when they are actually closer to the goal. Similarly for the Maze Navigation task, counting all obstacles in the sub-maze as part of the hardness cost is problematic, as obstacles outside the plan trajectory can inflate the difficulty inaccurately.
4. In Section 2.3, under Controller Training Data, Step 3, it’s unclear why only a contiguous chunk should be assigned to System-2. What is the rationale for restricting it to contiguous data?
5. According to the authors' definition of sub-goals, are they explicitly training the Controller to decompose the task into exactly three sub-goals?

**Minor Comments:**
1. Examples in the figures, particularly Figure 2, would benefit from clearer annotations in the maze to improve the interpretation of the numbering. The current numbering follows a top-to-bottom, left-to-right order, but this is not immediately clear.
2. In Figure 2, System-2 output, I do not understand why this is invalid “Exploring action left State [3, 1] is invalid”?
3. The authors should maintain consistency in their notation. In Section 2.3, when discussing Controller training data, the notation for initial and goal states switches from $s_i$​ and $s_j$​ in Step 1 to $s_0$​ and $s_g$​ later in the paragraph.


**References:**

[1] Fabiano, F., Pallagani, V., Ganapini, M.B., Horesh, L., Loreggia, A., Murugesan, K., Rossi, F. and Srivastava, B., 2023, December. Plan-SOFAI: A Neuro-Symbolic Planning Architecture. In Neuro-Symbolic Learning and Reasoning in the era of Large Language Models.

---

> ### Author Response · Authors · 2024-11-25
> **Response to Reviewer eYzS (Part 1)**
>
> Thank you for your review!
>
> > A significant limitation of this approach is its dependency on multiple components performing flawlessly
>
> The fact that System 1.x is built with three different modules should be seen as an advantage rather than a limitation. This means that one can directly swap in different versions of the System 1 Planner, the System 2 Planner, or the Controller and still retain the conceptualization of System 1.x, which, as we note, is our contribution. This is the reason why we start with general definitions of each of these components and experiment with different versions of the controller (by varying the hardness functions) and System-2 planners (LLM-based and A*). We want to emphasize that such variations are only possible because of the disentanglement of capabilities in our hybrid System 1.x. On the specific point about compounding errors, it’s always a possibility for any multi-model system; however, we show that our approach does better empirically than baselines. It remains to be seen how our method scales for arbitrarily hard planning tasks but that is beyond the scope of this paper.
>
> > A limitation of this hybrid approach is that selecting the x value too greedily can hurt performance and increase training computation costs.
>
> This, again, should be seen as an advantage of our method that users can control "x" based on their compute budget  and incidentally, also noted in one of your strengths – "This allows users to balance speed and accuracy". Naturally, there’s always going to be a trade-off between always doing deliberate search versus not doing so at all. However, the presence of a controller ensures that whatever compute budget is allowed, our system allocates them intelligently to solve the harder sub-goals.
>
> > For System-2, the hardness function should be explicitly defined for each specific domain in which the planner operates.
>
> First, note that System-2 does **not** rely on any hardness function. We fine-tune an LLM to carry out search by training it on search trajectories, which in our case, comes from running A* but could generally be curated in any other way (e.g., expert demonstrations from humans, etc).
>
> Next, if you meant that our controller requires defining hardness functions, we would like to clarify that is only for **automatically generating training data** (see that Algo 1 in the paper is only for training data construction). If a certain domain already has access to some easy/hard annotations of sub-goals (from a human or otherwise), the controller can be trained on that data alone without having to define any hardness function. However, since we may not always have access to such data, we show that it is also possible to automatically generate training data by using some domain knowledge, which in our case translates to hardness functions. Even such hardness functions can be quite flexible and need not be perfect because they are essentially heuristics for constructing training data – as we show in Table 10, System 1.x performs comparably for Mazes using either Obstacles or Manhattan Distance as the hardness function.
>
> Overall, we again want to stress the importance of System 1.x as a novel framework for hybrid and controllable planning with LLMs. The specifics of how we obtain training data to build the controller can vary, based on the end-goal/domain. We hope that future works that build on top of our framework will explore such possibilities.

---

> > ### Author Response · Authors · 2024-11-25
> > **Response to Reviewer eYzS (Part 2)**
> >
> > > The term plan validity should not be used interchangeably with plan accuracy.
> >
> > Yes, we meant validity. We will make this change in a future version.
> >
> > > The definition of a sub-goal as a "pair of states" adds unnecessary complexity and deviates from the usual understanding
> >
> > We are happy to make this change and will do so in a future version.
> >
> > > The hardness function defined for Blocksworld, in Section A.3, seems wrong.
> >
> > Hardness functions, as we define in our paper, are heuristics that can help us automatically label some training data. This means that they need not be perfect and can of course be improved upon. However, finding the best hardness function is not the primary focus of this paper. Moreover, we do not directly use these heuristics for hardness estimations. Instead, we fine-tune an LLM with such data and thus, also rely on its pre-trained knowledge to make these sub-goal/hardness predictions. For Mazes, we also experiment with Manhattan Distance in Table 10 and we show that it performs comparably to #Obstacles.
> >
> > > In Section 2.3, under Controller Training Data, Step 3, it’s unclear why only a contiguous chunk should be assigned to System-2
> >
> > Assigning System-2 to one contiguous chunk makes it easier to solve the constrained optimization problem in Line 261. Intuitively, it means that we are trying to annotate that **single** **hardest** sub-goal as System-2 such that the other (easy) sub-goals automatically get annotated as System-1.
> >
> > This, as asked in your follow-up question, indeed limits the maximum number of sub-goal decompositions to 3. It’s 1 when the problem is System1-only (Step 2 of the algo) or 2 when the System-2 sub-goal is on either end of the plan.
> >
> > In general, there could be more than 3 sub-goals and the number of System-2 sub-goals could also be more than one. However, since we are trying to automatically generate sub-goal training data (a hard task, in our opinion), as a starting point we made some simplifying assumptions which, if sub-goals annotations were to be readily available, would not be necessary.
> >
> > > In Figure 2, System-2 output, I do not understand why this is invalid “Exploring action left State [3, 1] is invalid”?
> >
> > Thanks for noting this, it’s a typo and should be "valid". To clarify, we mark a state as invalid if (1) it has an obstacle, (2) it’s outside the maze dimensions, and (3) it’s already visited. The Appendix has some detailed examples.
> >
> > > The authors should maintain consistency in their notation.
> >
> > We believe the notations are consistent but would be happy to check things further if you point to some other inconsistencies. Specifically, s_0 and s_g refer to the start and the goal state of the original problem while s_i and s_j in Line 239 refer to the start and the goal state of a particular sub-goal (since we defined sub-goals as a pair of states and hardness functions can be defined for any sub-goal).

---

> > ### Comment · Reviewer_eYzS · 2024-11-29
> > **Thank you**
> >
> > Thank you for your response, which has clarified a few of my points.
> >
> > Regd. Hardness function: Yes, I meant the **controller** requires hardness function estimates for it to be trained properly. Thank you for citing Table 10. I agree with the Reviewer **vcqi** suggestion of using standard heuristics for this process.

---

### Author Response · Authors · 2024-11-25
**General Response**

Based on the reviewer comments, there seems to be some misunderstanding about what System 1.x is and what it isn’t. In this general response, we aim to clarify some of those.

**1. System-1.x is about planning with LLMs**: System 1.x is a framework for hybrid and controllable planning, specifically **with LLMs**. Note the title of our paper and recall how all our components (System 1/2/controller) are fine-tuned LLMs from a single base architecture. This should help clarify the following:

- that our research goal is the following – **"How can we make LLMs better and more efficient planners, compared to pure System-2 methods that always conduct token-based search (e.g., Searchformer, Stream-of-Search) for all sub-goals?"**
- that our research goal is **not** the following – **"What is the best (neuro-symbolic) architecture for solving Mazes and Blocksworld?"**, which we know A* is already good at. However, that should not stop us from exploring whether LLMs can solve such problems too and to what extent and under what budget constraints, etc.
- that it should suffice to compare System 1.x Planner to an LLM-based System-1 Planner (baseline) and an LLM-based System-2 Planner (skyline). While there are many other architectures/methods for solving planning problems, they are not within the scope of the specific research question we are interested in.

**2. Neuro-symbolic System 1.x is only a variant**: Our paper, in some capacity, also talks about a neuro-symbolic System 1.x where the System-2 planner is A*. We do so for the sake of completeness and to show the flexibility of our modular design that allows one to swap in different versions of System 1/2/controller and still retain the conceptualization of our System-1.x. Hence, this modular design should be seen as an advantage and not a limitation.

**3. System-1.x uses hardness functions as heuristics only for automatically constructing training data for the controller**: Our hardness functions are heuristics that help us provide some supervision for training the LLM-based controller. Hence, they need not be perfect, can of course be improved upon (but is not the focus of this paper), and more generally, the fine-tuning process still leverages the LLM’s pre-trained knowledge to make these hardness/sub-goal predictions. If some human annotations of such sub-goals are available, one does not even need these hardness functions and can directly train the controller on that data.

---

### Meta-Review · Area_Chair_CAwX · 2024-12-23

**Metareview:**

This paper presents a framework for LLM-based planning that adaptively interleaves quick and cost-efficient planning with more costly but performant planning. This is achieved by having a controller decompose the problem into sub-goals and then classifying them as requiring the efficient or performant planner. The paper demonstrates that this framework creates a planner with improved controllability, flexibility, and generalizability.

**Additional Comments On Reviewer Discussion:**

The authors provided detailed responses to the concerns raised by the reviewers. However, there was only limited engagement from the reviewers during the rebuttal period. In some cases the author response clearly addresses the concerns raised in the review but is met with little to no acknowledgment.

Two reviewers strongly recommended rejection:
Reviewer z8RW did not engage with the authors or other reviewers during the rebuttal and discussion period despite recommending rejection. Furthermore, the authors provided a detailed response that addressed all the weaknesses/questions raised in the review. I therefore feel confident about disregarding z8RW’s rejection recommendation.

Reviewer s6YJ, raised some interesting questions about the paper and I thank the reviewer for engaging with the authors and other reviewers during the discussion period. However, I strongly disagree with the claim that the evaluation was done in bad faith (such a claim should not be made without clear evidence of malicious intent). From the review and discussion it seems that the reviewer disagrees with the central premise of building a LLM-based planning system. However, this is more of a research position and there were insufficient concrete concerns raised in the review about the paper. The two main outstanding concerns, namely (i) bad labelling of an axis in one plot, and (ii) insufficient discussion about A*, are not significant enough to warrant rejection.

Reviewers vcqi and eYzS have given this paper borderline scores. Having carefully reviewed the discussion and the paper, I agree with the authors’ viewpoint that the chief concern about heuristics would not impact the research questions of the paper. While the reviewers have a valid suggestion, that using standard heuristics would address a frequent pain-point, this would be a relatively minor improvement to the framework in terms of research novelty and the experiments already show evidence of the usefulness of the proposed technique.

Overall, I thank all the reviewers for their detailed reviews and urge the authors to make the changes that they have committed to making in their responses, for the camera-ready version of the paper. Despite those shortcomings, the paper makes a significant scientific contribution which meets the threshold for acceptance.

---

### Decision · Program_Chairs · 2025-01-22

Accept (Poster)